# SEGA: Shaping Semantic Geometry for Robust Hashing under Noisy Supervision

**Yiyang Gu**[1*]**, Bohan Wu**[1*]**, Qinghua Ran**[2]**, Rong-Cheng Tu**[3]**, Xiao Luo**[4†]**,**

**Zhiping Xiao**[5†]**, Wei Ju**[1]**, Dacheng Tao**[3]**, Ming Zhang**[1†]

[1] State Key Laboratory for Multimedia Information Processing,
School of Computer Science, PKU-Anker LLM Lab, Peking University
[2] School of Mathematical Sciences, Peking University
[3] Nanyang Technological University [4] UCLA [5] University of Washington
{yiyanggu,wxtpku,juwei,mzhang_cs}@pku.edu.cn,
rqh143@stu.pku.edu.cn,
{rongcheng.tu,dacheng.tao}@ntu.edu.sg,
xiaoluo@cs.ucla.edu, patxiao@uw.edu

## Abstract

This paper studies the problem of learning hash codes from noisy supervision, which is a practical yet challenging task. This problem is important in extensive real-world applications such as image retrieval and cross-modal retrieval. However, most of the existing methods focus on label denoising to address this problem, but ignore the geometric structure of the hash space, which is critical for learning stable hash codes. Towards this end, this paper proposes a novel framework named Semantic Geometry Shaping (SEGA) that explicitly refines the semantic geometry of hash space. Specifically, we first learn dynamic class prototypes as semantic anchors and cluster hash embeddings around these prototypes to keep structural stability. We then leverage both the energy of predicted distributions and structure-based divergence to estimate the uncertainty of instances and calibrate the supervision in a soft manner. Moreover, we introduce structure-aware interpolation to improve the class boundaries. To verify the effectiveness of our design, we give the theoretical analysis for the proposed framework. Experiments on a range of widely-used retrieval datasets justify the superiority of our SEGA over extensive strong baselines under noisy supervision.

## 1 Introduction

Deep hashing methods are significant for large-scale and efficient information retrieval. The goal of these approaches is to generate compact binary codes retaining good semantic similarity [56]. In the past few years, supervised deep hashing has performed well in various retrieval tasks, such as image retrieval and multi-modal retrieval [34, 4, 35, 69, 32]. However, label noise is still an outstanding challenge when it comes to practical applications [50, 20, 31]. In real-world datasets, labels are often incomplete, ambiguous, or incorrect [68, 60, 18]. These problems may stem from inconsistent human annotations, noisy web collection, loose category definitions, and intrinsic semantic ambiguity [23, 65]. The discrete property of hash codes makes this problem worse. Hashing models are sensitive to noisy supervision, where small label errors can cause large output shifts. This leads to fragmented representations and unstable similarity preservation [47]. When supervision

---

[*]Equal contribution.
[†]Corresponding authors.

39th Conference on Neural Information Processing Systems (NeurIPS 2025).

is unreliable, models that fit labels directly tend to overfit noise [64, 14] instead of learning real class structure [1, 38]. The main challenge in noisy multi-label hashing is not only to maintain class separability under weak signals but also to build an embedding space that is semantically consistent, geometrically smooth, and robust to supervision noise.

In order to alleviate this problem, previous studies mainly focused on denoising labels or regularizing predictions. These include approaches that reweight or clear training samples [39, 7], estimate noise transition matrices [43], apply auxiliary consistency constraints [68], or perform confidence-based sample selection [20, 31]. These strategies have been proven to be effective in reducing the impact of noisy supervision, mainly by improving the quality of labels or learning stability. However, they basically follow a label-centric paradigm, assuming the label, whether observed or corrected, remains the main source of supervision. In contrast, we propose a complementary perspective that focuses on modeling and reinforcing the semantic geometry of the representation space. The goal of deep hashing is to encode the inputs into discrete codes that reflect semantic similarity. The learned hash codes may be suboptimal when hash embeddings fail to align with the underlying semantic manifolds. In contrast, the semantical and topological consistency of the hash space can make the hash codes more robust against label noise. It raises two key challenge to build such a geometry-aware framework for robust hash learning: *(1) How can we evaluate the uncertainty of supervision accurately in the presence of label noise?* It is difficult to determine the uncertainty of supervised signals from traditional confidence measures under heavy label noise. However, we can capture additional cues of uncertainty from the geometric structure of representations. *(2) How to capture the underlying semantic structure of data effectively under noisy multi-label supervision?* It is promising to shape the semantic geometry of hash space to align the semantic relationship among the instances.

Towards this end, this paper proposes a novel framework named Semantic Geometry Shaping (SEGA) that explicitly refines the semantic geometry of hash space. It integrates three mutually promoting components into a closed-loop training system. First, *Prototype-Guided Semantic Anchoring* aligns instance embeddings with dynamic class prototypes that evolve during training, which serve as structural anchors. Second, *Uncertainty-Guided Supervision Calibration* calculates a unified uncertainty score by combining energy-based prediction confidence and structure-based divergence, allowing soft weighting of noisy labels. Third, *Structure-Aware Mixup* interpolates between samples with different uncertainties but similar semantics to refine ambiguous regions, so as to promote continuity and improve decision boundaries. These modules enable representations, supervision, and uncertainty to coevolve in a tightly coupled geometric loop. We have verified SEGA from both theoretical and empirical aspects, and the results show that it can approximately evaluate the reliability of labels to achieve supervised weighting, and maintain semantic consistency in the interpolation process, thus reducing structural fragmentation and regularizing decision boundaries under noise. Experiments on noisy multi-label hashing benchmarks show that SEGA achieves state-of-the-art robustness, particularly under extreme label corruption. In addition to numerical improvement, the visualization results also confirm that SEGA can construct discriminative and semantically coherent representations under noisy supervision. These findings suggest that shaping semantic geometry, rather than merely correcting supervision, is crucial for resilient hash algorithms.

Our main contributions are as follows. (1) *New Perspective*. We present a geometric view of learning under noisy supervision. The multi-label hashing problem is reformulated as semantic structure alignment rather than simple label recovery. (2) *Novel Methodology.* We propose SEGA, a closed-loop framework that unifies prototype anchoring, uncertainty-guided soft labeling, and structure-aware interpolation into a geometry-aware semantic learning process. (3) *Extensive Experiments.* We evaluate SEGA on four benchmark datasets. It consistently achieves state-of-the-art performance and shows strong robustness across diverse noisy multi-label settings.

## 2   Related Work

**Deep Hashing Methods.** Deep hashing has become a core approach for scalable similarity retrieval. It learns compact binary codes that preserve semantic relations in Hamming space. Supervised methods usually build pointwise, pairwise, or triplewise objectives based on label-derived similarity [34, 69, 4, 32]. For instance, GreedyHash [51] utilizes the greedy strategy to address the gradient-vanishing problem in discrete hashing optimization. CSQ [63] proposes a global central similarity metric to improve hash learning efficiency. Unsupervised hashing methods usually leverage the intrinsic relationships in data to avoid label dependence. For example, WCH [62] introduces

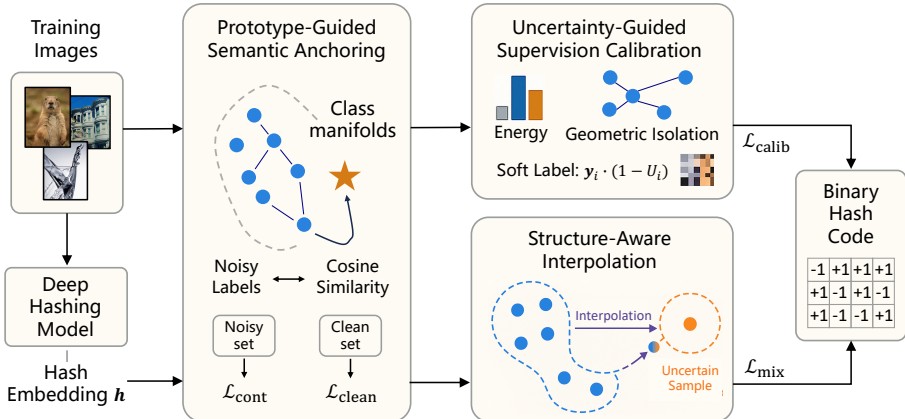

Figure 1: An overview of the proposed framework SEGA for robust multi-label hashing under noisy supervision. It integrates prototype anchoring, uncertainty-calibrated soft labels, and structure-aware interpolation into a closed-loop system that aligns semantic supervision with representation geometry.

weighted contrastive learning and mutual attention mechanism to enhance data similarity mining. These methods perform poorly when the provided supervision contains noise, as they are unable to effectively utilize the useful signals within the noisy supervision or reduce the interference caused by the noise. Recent efforts toward noise-robust hashing include DIOR [55] and STAR [41]. DIOR utilizes a dual-partition strategy to relieve the impact of noisy labels. STAR leverages a hybrid sample selection mechanism and selective centroid learning to help capture similarity structure. however, they mainly focus on label denoising and local noise reduction, but overlook the structural fragility of the global hash space. Towards this end, we propose a novel framework that explicitly refines the semantic geometry of hash space to alleviate the influence of label noise.

**Learning with Label Noise.** Label noise is common in real-world datasets and has motivated extensive researches on robust learning against noise [50, 26, 17]. The existing methods can be mainly divided into three categories. The first category of methods refines losses by modeling noise transformation [43, 15, 45]. The second one leverages sample selection to identify clean data [20, 31, 68]. The third one utilizes regularization to alleviate memorization of noisy labels [66, 21, 57]. Co-teaching [20] introduced a dual-network design that filters noisy samples through cross-selection of small-loss instances. Mixup [66] provides implicit regularization by linearly interpolating samples and labels, reducing overfitting to corrupted data. Recent advances combine contrastive learning [9, 27] with noise-robust representations [13, 61] and apply curriculum learning to control memorization dynamics [1]. Early learning regularization [38] limits overfitting by focusing on clean patterns in early training. Despite this progress in image classification, robust learning for structured outputs such as hashing remains less explored, especially under ambiguous multi-label supervision.

## 3 Methodology

### 3.1 Problem Definition

Let $\mathcal{D} = \{(\boldsymbol{x}_i, \boldsymbol{y}_i)\}_{i=1}^N$ be a training set of $N$ examples, where $\boldsymbol{x}_i \in \mathbb{R}^d$ is an input instance and $\boldsymbol{y}_i \in \{0,1\}^C$ is a one-hot or multi-hot label vector over $C$ semantic categories. In practical settings, these labels are often noisy due to annotation errors, class ambiguity, or missing tags. The goal is to learn a hash function $H : \boldsymbol{x} \mapsto \boldsymbol{b} \in \{-1, +1\}^L$ that encodes each sample into an $L$-bit binary code, such that semantically similar inputs are mapped to nearby codes in the Hamming space.

### 3.2 Framework Overview

SEGA addresses this goal by shaping a robust semantic geometry in the embedding space, even under noisy multi-label supervision. We adopt a deep hashing model $f : \boldsymbol{x} \mapsto \boldsymbol{h} \in \mathbb{R}^L$ to produce continuous hash embeddings, which are binarized as $\boldsymbol{b} = \text{sign}(\boldsymbol{h})$. However, directly training on corrupted labels $\boldsymbol{y}_i$ often induces semantic misalignment, especially when representations are influenced by unreliable or ambiguous supervision. To address this problem, SEGA shapes the semantic geometry of the representation space with three key parts. First, *prototype-guided semantic*

*anchoring* aligns embeddings with class prototypes to keep the geometry stable. Second, *uncertainty-guided supervision calibration* reduces the weight of unreliable labels using energy and structural divergence. Third, *structure-aware interpolation* mixes only semantically similar samples with different uncertainty levels to create smooth transitions near class boundaries. The components form a closed loop. Supervision, uncertainty, and geometry co-evolve and reinforce each other through training. The overview of the proposed framework SEGA is illustrated in Figure 1.

### 3.3 Prototype-Guided Semantic Anchoring

SEGA builds a semantic scaffold in the hash space using a set of learnable class prototypes $\{\boldsymbol{p}_c\}_{c=1}^C$. Each prototype $\boldsymbol{p}_c \in \mathbb{R}^L$ is randomly initialized and updated together with the model. These prototypes act as anchors that guide instance embeddings toward meaningful semantic directions. For each input $\boldsymbol{x}_i$, we obtain its continuous hash representation $\boldsymbol{h}_i = f(\boldsymbol{x}_i)$ and normalize it as $\hat{\boldsymbol{h}}_i = \boldsymbol{h}_i / \|\boldsymbol{h}_i\|$. Similarly, we normalize all prototypes to get unit-length semantic anchors: $\hat{\boldsymbol{p}}_c = \boldsymbol{p}_c / \|\boldsymbol{p}_c\|$. The logit vector for classification is then computed as:

$$\boldsymbol{l}_i = \hat{\boldsymbol{h}}_i^\top [\hat{\boldsymbol{p}}_1, \hat{\boldsymbol{p}}_2, \ldots, \hat{\boldsymbol{p}}_C], \tag{1}$$

which yields cosine similarities between the embedding and each class prototype. To assess the semantic reliability of supervision, we define a soft-alignment score that evaluates the agreement between model prediction and the observed (possibly noisy) label vector $\boldsymbol{y}_i$:

$$S_r^{(i)} = \cos\left(\sigma(\boldsymbol{l}_i), \ \boldsymbol{y}_i\right), \tag{2}$$

where $\sigma(\cdot)$ denotes the softmax function. This score measures how well the model's predicted distribution aligns with the noisy label in the prototype-induced space. Based on the $q_r$-th percentile of the alignment scores $\{S_r^{(i)}\}_{i=1}^N$, we compute a dynamic threshold $\tau_r$ to partition the dataset into a clean set $\mathcal{D}_c$, containing semantically reliable samples, and a complementary noisy set $\mathcal{D}_n$:

$$\tau_r = \text{Percentile}(\{S_r^{(i)}\}, q_r), \quad \mathcal{D}_c = \left\{\boldsymbol{x}_i \mid S_r^{(i)} \geq \tau_r\right\}, \quad \mathcal{D}_n = \mathcal{D} \setminus \mathcal{D}_c. \tag{3}$$

Clean samples in $\mathcal{D}_c$ are supervised using a softmax-based prototype alignment loss [55] that treats the noisy label vector as a probabilistic guide:

$$\mathcal{L}_{\text{clean}} = - \sum_{\boldsymbol{x}_i \in \mathcal{D}_c} \sum_{k=1}^C \frac{\boldsymbol{y}_{ik}}{\sum_{k=1}^C \boldsymbol{y}_{ik}} \log\left(\frac{\exp(\hat{\boldsymbol{p}}_k^\top \hat{\boldsymbol{h}}_i)}{\sum_{j=1}^C \exp(\hat{\boldsymbol{p}}_j^\top \hat{\boldsymbol{h}}_i)}\right). \tag{4}$$

To regularize noisy samples in $\mathcal{D}_n$ and prevent semantic drift, we adopt a contrastive loss [46, 27, 54]. It pulls embeddings of related samples closer and pushes unrelated ones apart by a margin $m$. For two samples $\boldsymbol{x}_i, \boldsymbol{x}_j \in \mathcal{D}_n$ with noisy labels $\boldsymbol{y}_i$ and $\boldsymbol{y}_j$, we define:

$$\mathcal{L}_{\text{cont}} = \frac{1}{|\mathcal{D}_n|^2} \sum_{\boldsymbol{x}_i, \boldsymbol{x}_j \in \mathcal{D}_n} \left[\mathbb{I}_{[i \sim j]} \cdot \|\hat{\boldsymbol{h}}_i - \hat{\boldsymbol{h}}_j\|^2 + \mathbb{I}_{[i \not\sim j]} \cdot \max(0, m - \|\hat{\boldsymbol{h}}_i - \hat{\boldsymbol{h}}_j\|)^2\right], \tag{5}$$

where $\mathbb{I}_{[i \sim j]} = 1$ if $\boldsymbol{y}_i^\top \boldsymbol{y}_j > 0$ (i.e., positive pair), and $\mathbb{I}_{[i \not\sim j]} = 1$ otherwise (i.e., negative pair). Prototype-guided alignment provides reliable supervision. Contrastive regularization stabilizes learning in noisy regions. Together, they help build a structured representation space that preserves semantic geometry under label noise.

### 3.4 Uncertainty-Guided Supervision Calibration

The previous module divides the dataset into clean and noisy parts using prototype alignment. This hard split ignores the reliability differences within each part. Some clean samples lie near decision boundaries or show structural ambiguity. Some noisy samples still contain useful semantic information. To handle this, we introduce a novel metric to evaluate uncertainty [44, 11] of samples and then calibrate the supervision in a soft manner. Specifically, we combine energy-based prediction confidence and structure-based neighborhood divergence to estimate an uncertainty score. We then leverage this score to reduce the weight of supervision for ambiguous samples and filter out unreliable gradients. It also guides later sample selection in structure-aware interpolation.

**Energy-based prediction confidence.** We measure the confidence of the model using the energy [40, 67] of its logit distribution:

$$E_i = -\log \sum_{c=1}^{C} \exp(\boldsymbol{l}_{ic}). \tag{6}$$

A high energy value means that the distribution is flat and the prediction is less certain. We normalize the energy in each mini-batch as $\tilde{E}_i = \frac{E_i - \min_j E_j}{\max_j E_j - \min_j E_j}$, so that all values stay on a comparable scale.

**Structure-based divergence.** Confidence alone cannot reflect spatial consistency in the embedding space. We therefore define a divergence score. It measures how far a sample spreads from its $K$ nearest neighbors in geometry:

$$\delta_i = 1 - \frac{1}{K} \sum_{j \in N_i} \cos(\hat{\boldsymbol{h}}_i, \hat{\boldsymbol{h}}_j). \tag{7}$$

Here, $N_i$ is the set of top-$K$ nearest neighbors for sample $\boldsymbol{x}_i$. A larger $\delta_i$ means the sample is more isolated in geometry. Such samples often lie near class boundaries or in under-represented semantic zones.

**Unified uncertainty.** The total uncertainty is defined as:

$$U_i = (1 - \tilde{E}_i) \cdot \delta_i. \tag{8}$$

This design captures two kinds of cues: vertical cue from prediction confidence and lateral cue from structural smoothness. It gives a unified measure of semantic reliability. Instead of discarding uncertain samples, we calibrate their gradient impact by softly reweighting the label signal [42, 29]:

$$\mathcal{L}_{\text{calib}} = -\frac{1}{|\mathcal{D}|} \sum_{\boldsymbol{x}_i \in \mathcal{D}} \sum_{c=1}^{C} \tilde{\boldsymbol{y}}_{ic} \cdot \log[\sigma(\boldsymbol{l}_i)]_c, \quad \tilde{\boldsymbol{y}}_i = \boldsymbol{y}_i \cdot (1 - U_i), \tag{9}$$

where $\sigma(\cdot)$ is the softmax function. This calibration reduces the effect of samples that are semantically or structurally uncertain. It improves the shape of semantic manifolds and strengthens alignment inside coherent regions. The resulting gradients mainly update well-supported semantic areas and smooth the uncertain ones without forcing hard boundaries.

**Theoretical Analysis.** We now give a theoretical explanation for our uncertainty-guided supervision calibration. The uncertainty score $U_i$ combines semantic confidence and structural consistency. It approximates the posterior probability that the observed label is correct. With a mild conditional independence assumption, $1 - U_i$ can be used as a proper weight in the supervision loss.

**Theorem 3.1** (Uncertainty-Weighted Confidence Approximates Label Correctness). *Let $\boldsymbol{x}_i$ be a training sample with observed label $\tilde{\boldsymbol{y}}_i$ and true (latent) label $\boldsymbol{y}_i^*$. Define the energy-based prediction confidence as $E_i = -\log \sum_{c=1}^{C} \exp(\boldsymbol{l}_{ic})$, and $\tilde{E}_i = \frac{E_i - \min_j E_j}{\max_j E_j - \min_j E_j}$, the structure-base divergence as $\delta_i = 1 - \frac{1}{K} \sum_{j \in \mathcal{N}_i} \cos(\hat{\boldsymbol{h}}_i, \hat{\boldsymbol{h}}_j)$. If we assume the probability that the label is correct depends solely on the semantic and structural reliability of the sample, and these two sources are conditionally independent given $\boldsymbol{x_i}$, then:*

$$\Pr[\boldsymbol{y}_i^* = \tilde{\boldsymbol{y}}_i \mid \boldsymbol{x}_i] \propto 1 - U_i. \tag{10}$$

This result provides a probabilistic interpretation of $1 - U_i$ as an estimate of label correctness. By treating semantic confidence and structural consistency as two independent indicators of supervision quality, $1 - U_i$ serves as a soft surrogate for the reliability of the observed label. Therefore, using $1 - U_i$ to weight the label signal helps the model reduce gradients from noisy or ambiguous samples. It keeps stronger supervision for reliable regions. This process aligns the learning dynamics with semantically meaningful structures.

## 3.5 Structure-Aware Interpolation for Boundary Regularization

Uncertainty-guided calibration reduces noisy gradients but does not handle semantic ambiguity near decision boundaries. Samples in these areas are often uncertain and structurally isolated. To solve this, we draw inspiration from Mixup [53, 59, 37], which smooths class transitions and

makes representations more generalizable by mixing samples. Based on this idea, we design a structure-aware interpolation mechanism for refining boundaries in noisy multi-label hashing.

Traditional Mixup mixes any two samples without restriction. In our method, we mix only pairs that are semantically similar but have different uncertainty levels. We split the clean set $\mathcal{D}_c$ into two parts: a confident subset $\mathcal{D}_{\text{con}}$ and an uncertain subset $\mathcal{D}_{\text{un}}$. The split is based on the mean uncertainty $\bar{U}$:

$$\mathcal{D}_{\text{con}} = \{\boldsymbol{x}_i \in \mathcal{D}_c \mid U_i \leq \bar{U}\}, \quad \mathcal{D}_{\text{un}} = \{\boldsymbol{x}_i \in \mathcal{D}_c \mid U_i > \bar{U}\}. \tag{11}$$

For each uncertain sample $\boldsymbol{x}_i \in \mathcal{D}_{\text{un}}$, we find its most similar confident sample $\boldsymbol{x}_j^* \in \mathcal{D}_{\text{con}}$. The two samples must share at least one common label, that is, $\boldsymbol{y}_i^\top \boldsymbol{y}_j > 0$. We measure similarity by cosine distance in the normalized embedding space:

$$\boldsymbol{x}_j^* = \arg \max_{\boldsymbol{x}_j \in \mathcal{D}_{\text{con}}, \, \boldsymbol{y}_i^\top \boldsymbol{y}_j > 0} \cos(\hat{\boldsymbol{h}}_i, \hat{\boldsymbol{h}}_j). \tag{12}$$

This rule keeps interpolation between samples that share some labels. It preserves semantic consistency and uses confident samples as anchors to guide uncertain ones. We then generate interpolated virtual samples and soft labels as:

$$\tilde{\boldsymbol{x}} = \lambda \boldsymbol{x}_i + (1 - \lambda)\boldsymbol{x}_j^*, \quad \tilde{\boldsymbol{y}} = \lambda \tilde{\boldsymbol{y}}_i + (1 - \lambda)\tilde{\boldsymbol{y}}_j^*, \quad \lambda \sim \text{Beta}(\alpha, \alpha). \tag{13}$$

Here, $\tilde{\boldsymbol{y}}_i$ and $\tilde{\boldsymbol{y}}_j^*$ are the uncertainty-calibrated soft labels described in Section 3.4. The coefficient $\lambda$ comes from a symmetric Beta distribution to provide diverse mixing strengths. We define the mixup loss as follows:

$$\mathcal{L}_{\text{mix}} = -\frac{1}{|\mathcal{D}_{\text{mix}}|} \sum_{(\tilde{\boldsymbol{x}}, \tilde{\boldsymbol{y}}) \in \mathcal{D}_{\text{mix}}} \sum_{c=1}^{C} \tilde{\boldsymbol{y}}_c \cdot \log[\sigma(f(\tilde{\boldsymbol{x}}))]_c, \tag{14}$$

where $f(\cdot)$ is the hash encoder and $\sigma(\cdot)$ denotes the softmax function. This structure-aware interpolation transmits semantics and supervision signals from confident samples to uncertain samples, which introduces additional regularity near the decision boundaries.

**Theoretical Analysis.** We provide a theoretical analysis to justify the effectiveness of structure-aware interpolation. It demonstrates that the interpolation keeps the consistency of the semantic structure in the hash space.

**Theorem 3.2** (Structure-Preserving Interpolation Bound). *Let $\boldsymbol{x}_i$ and $\boldsymbol{x}_j$ be two clean samples belonging to class c, i.e., $\boldsymbol{y}_{ic} = \boldsymbol{y}_{jc} = 1$. $\hat{\boldsymbol{h}}_i$ and $\hat{\boldsymbol{h}}_j$ represent the normalized hash embeddings of these two samples. Let $\boldsymbol{p}_c$ be the unit-norm prototype vector of class c. For any $\lambda \in [0, 1]$, the interpolated embedding is defined as follows:*

$$\tilde{\boldsymbol{h}} = \lambda \hat{\boldsymbol{h}}_i + (1 - \lambda)\hat{\boldsymbol{h}}_j$$

*The cosine similarity between $\tilde{\boldsymbol{h}}$ and $\boldsymbol{p}_c$ has the following lower bound:*

$$\cos(\tilde{\boldsymbol{h}}, \boldsymbol{p}_c) \geq \lambda \cos(\hat{\boldsymbol{h}}_i, \boldsymbol{p}_c) + (1 - \lambda) \cos(\hat{\boldsymbol{h}}_j, \boldsymbol{p}_c)$$

This result indicates that structure-aware interpolation preserves the proximity to the class prototype along the interpolation path. It enhances the consistency of the semantic structure in uncertain regions and improves the class boundaries.

## 3.6 Unified Training Objective

We integrate all modules into a unified training objective. It includes four complementary loss terms that jointly learn semantic anchors, maintain structural stability, calibrate noisy supervision, and improve class boundaries against label noise.

$$\mathcal{L} = \mathcal{L}_{\text{clean}} + \mathcal{L}_{\text{cont}} + \mathcal{L}_{\text{calib}} + \mathcal{L}_{\text{mix}}. \tag{15}$$

Each loss term contributes to shaping the semantic geometry of the hash space differently: (1) $\mathcal{L}_{\text{clean}}$ clusters clean samples around their corresponding prototypes, which ensures stable class semantics; (2) $\mathcal{L}_{\text{cont}}$ enhances the structural consistency for noisy instances; (3) $\mathcal{L}_{\text{calib}}$ alleviates the impact of unreliable supervision using both the energy of predicted distributions and structure-based divergence; (4) $\mathcal{L}_{\text{mix}}$ facilitates the consistency of the semantic structure in uncertain regions and regularizes the class boundaries. It makes the learned hash codes more robust and generalizable to optimize the total loss objective. The overall training algorithm of our proposed SEGA is provided in Algorithm 1.

**Algorithm 1:** Training Procedure of SEGA

| | |
|---|---|
| **Input** | : Noisy dataset $\mathcal{D}$; encoder $f(\cdot)$ with parameters $\Theta$; prototypes $\{\boldsymbol{p}_c\}_{c=1}^C$ |
| **Hyperparameters** | : Code length $L$; Mixup coefficient $\alpha$; partition percentile $q_r$ |
| **Output** | : Trained parameters $\Theta$; learned prototypes $\{\boldsymbol{p}_c\}_{c=1}^C$ |

Initialize $\boldsymbol{p}_c \sim \mathcal{N}(\boldsymbol{0}, \mathbf{I})$ for $c = 1, \ldots, C$ ;
**while** *not converged* **do**

> Compute embeddings $\hat{\boldsymbol{h}}_i = f(\boldsymbol{x}_i)$ and logits $\boldsymbol{l}_i$ via Eq. 1 ;
>
> Compute alignment scores $S_r^{(i)}$ via Eq. 2 and partition $\mathcal{D} \to \mathcal{D}_c, \mathcal{D}_n$ using Eq. 3 ;
> Compute $\mathcal{L}_{\text{clean}}$ on $\mathcal{D}_c$ via Eq. 4 and $\mathcal{L}_{\text{cont}}$ on $\mathcal{D}_n$ via Eq. 5 ;
> For each $\boldsymbol{x}_i \in \mathcal{D}$, compute $U_i$ via Eqs. 6-8 ;
> Calibrate labels $\tilde{\boldsymbol{y}}_i = \boldsymbol{y}_i \cdot (1 - U_i)$ and compute $\mathcal{L}_{\text{calib}}$ via Eq. 9 ;
> Partition $\mathcal{D}_c \to \mathcal{D}_{\text{con}}, \mathcal{D}_{\text{un}}$ by $\bar{U}$ via Eq. 11 ;
> For each $\boldsymbol{x}_i \in \mathcal{D}_{\text{un}}$, select $\boldsymbol{x}_j^*$ via Eq. 12 ;
> Generate $(\tilde{\boldsymbol{x}}, \tilde{\boldsymbol{y}})$ via Eq. 13 and compute $\mathcal{L}_{\text{mix}}$ via Eq. 14 ;
> Update $\Theta, \{\boldsymbol{p}_c\}_{c=1}^C$ by minimizing total loss Eq. 15.

Table 1: The comparison of MAP scores on four datasets under pairflip label noise.

| Method | CIFAR-10 | | | | FLICKR25K | | | | NUS-WIDE | | | | MS COCO | | | |
|---|---|---|---|---|---|---|---|---|---|---|---|---|---|---|---|---|
| Bits | 16 | 32 | 64 | 128 | 16 | 32 | 64 | 128 | 16 | 32 | 64 | 128 | 16 | 32 | 64 | 128 |
| DPSH | 51.89 | 54.04 | 54.41 | 56.32 | 58.29 | 59.72 | 59.89 | 59.65 | 42.98 | 44.32 | 45.74 | 46.17 | 32.69 | 33.47 | 34.07 | 34.33 |
| HashNet | 44.27 | 45.03 | 46.96 | 47.20 | 56.83 | 57.77 | 58.34 | 60.63 | 43.61 | 44.17 | 45.21 | 47.01 | 31.93 | 32.09 | 32.58 | 36.17 |
| SPQ | 41.11 | 42.35 | 43.87 | 45.67 | 52.75 | 53.21 | 54.18 | 54.56 | 40.03 | 41.20 | 41.75 | 42.31 | 27.98 | 28.35 | 28.97 | 30.17 |
| DCH | 54.17 | 54.28 | 55.27 | 57.17 | 58.87 | 59.54 | 60.20 | 60.09 | 43.31 | 44.39 | 45.69 | 46.29 | 34.52 | 35.15 | 35.81 | 36.09 |
| FSDH | 53.89 | 54.55 | 55.26 | 56.89 | 57.77 | 58.59 | 59.60 | 60.04 | 43.28 | 44.31 | 44.52 | 44.87 | 34.08 | 34.39 | 35.12 | 35.37 |
| GreedyHash | 50.97 | 52.14 | 53.41 | 54.18 | 57.98 | 58.04 | 59.05 | 59.77 | 43.78 | 44.26 | 44.49 | 44.69 | 34.17 | 34.21 | 34.31 | 34.66 |
| JMLH | 49.93 | 51.58 | 54.48 | 56.01 | 58.54 | 58.36 | 58.61 | 58.71 | 43.15 | 43.89 | 44.81 | 45.14 | 34.18 | 34.22 | 34.39 | 34.81 |
| DPN | 49.77 | 51.02 | 54.12 | 56.21 | 59.23 | 60.12 | 59.98 | 60.54 | 44.02 | 44.23 | 45.34 | 45.98 | 34.23 | 35.01 | 35.77 | 36.21 |
| WGLHH | 52.57 | 53.18 | 55.18 | 56.43 | 60.02 | 59.10 | 59.72 | 60.27 | 43.79 | 46.08 | 46.29 | 46.56 | 35.33 | 35.14 | 36.32 | 36.87 |
| CSQ | 53.13 | 54.27 | 55.33 | 58.25 | 59.47 | 60.02 | 60.38 | 61.37 | 43.89 | 43.67 | 44.92 | 46.13 | 34.27 | 34.12 | 34.60 | 34.78 |
| OrH | 54.13 | 55.15 | 57.67 | 58.37 | 58.39 | 60.29 | 59.54 | 60.19 | 44.44 | 44.74 | 44.45 | 46.61 | 34.84 | 34.90 | 34.88 | 34.97 |
| REL | 56.19 | 58.58 | 60.99 | 61.74 | 59.76 | 61.27 | 61.46 | 61.96 | 44.76 | 45.02 | 45.96 | 46.21 | 34.85 | 34.97 | 35.04 | 35.12 |
| Jo-SRC | 55.75 | 57.43 | 59.91 | 60.12 | 60.12 | 60.59 | 60.42 | 61.27 | 44.97 | 45.36 | 46.12 | 47.78 | 35.12 | 35.46 | 35.79 | 35.67 |
| DIOR | 60.96 | 67.89 | 69.36 | 71.17 | 64.36 | 65.44 | 66.78 | 68.21 | 50.02 | 51.40 | 52.26 | 52.64 | 37.14 | 38.22 | 38.78 | 38.92 |
| STAR | 69.05 | 69.82 | 70.52 | 71.83 | 70.54 | 70.76 | 71.40 | 71.96 | 56.15 | 56.77 | 57.69 | 58.17 | 41.72 | 42.04 | 42.86 | 43.21 |
| **SEGA (Ours)** | **70.01** | **70.59** | **72.34** | **73.56** | **71.19** | **72.14** | **74.49** | **76.48** | **59.11** | **59.52** | **61.86** | **64.61** | **44.37** | **46.08** | **47.21** | **47.58** |

## 4 Experiments

### 4.1 Setup

**Datasets.** We evaluate SEGA on four widely-used image retrieval benchmarks: CIFAR-10 [28], Flickr25k [24], NUS-WIDE [10], and MS COCO [36]. CIFAR-10 is a balanced single-label dataset with 10 classes of images. Flickr25k and NUS-WIDE are multi-label web datasets with 38 and 81 categories, respectively; we follow prior work [55] by selecting the top 10 classes in NUS-WIDE. MS COCO provides multi-object annotations over 80 classes with dense labels per image. All datasets come with clean annotations. To simulate realistic training noise, we inject synthetic corruption into a portion of training labels using two schemes: symmetric and pairwise noise. Noise rates are varied from 20% to 80% in steps of 20%. Symmetric noise randomly replaces each label with any other class, while pairwise noise flips labels only to semantically related categories [2].

**Baselines and metrics.** We compare SEGA with a diverse set of baselines grouped into three categories: (1) *Standard Deep Hashing Methods*, including DPSH [33], HashNet [5], SPQ [25], DCH [6], FSDH [19], GreedyHash [51], JMLH [48], DPN [12], WGLHH [52], CSQ [63], and OrH [22]; (2) *General Label Noise-Robust Methods*, including REL [57] and Jo-SRC [61]; and (3) *Noise-Resilient Hashing Methods*, including DIOR [55] and STAR [41]. We evaluate all methods at four code lengths: 16, 32, 64, and 128 bits. Performance is measured using the standard retrieval metric, i.e., mean average precision (MAP).

**Implementation Details.** All experiments are finished using PyTorch on a NVIDIA A40 GPU. We utilize stochastic gradient descent (SGD) with a momentum of 0.9 and a batch size of 24. The learning rate is initialized at 0.001, with weight decay set to 0.0004 and dropout rate to 0.5. The backbone network is initialized from a pretrained VGG-16 [49] model, consistent with all baseline

Table 2: The comparison of MAP scores on four datasets under symmetric label noise.

| Method | CIFAR-10 | | | | FLICKR25K | | | | NUS-WIDE | | | | MS COCO | | | |
|---|---|---|---|---|---|---|---|---|---|---|---|---|---|---|---|---|
| Bits | 16 | 32 | 64 | 128 | 16 | 32 | 64 | 128 | 16 | 32 | 64 | 128 | 16 | 32 | 64 | 128 |
| DPSH | 45.36 | 47.35 | 48.27 | 49.06 | 56.67 | 57.09 | 57.86 | 58.24 | 44.03 | 44.82 | 45.60 | 45.97 | 30.29 | 31.21 | 31.87 | 32.46 |
| HashNet | 42.54 | 43.24 | 44.47 | 45.29 | 53.47 | 54.69 | 56.03 | 56.87 | 43.21 | 44.37 | 45.12 | 46.05 | 27.17 | 27.69 | 28.54 | 28.89 |
| DCH | 45.29 | 47.75 | 48.19 | 49.24 | 57.02 | 58.20 | 58.44 | 59.12 | 44.36 | 44.21 | 44.98 | 45.88 | 31.55 | 32.39 | 32.99 | 33.41 |
| GreedyHash | 42.73 | 45.96 | 47.51 | 49.81 | 56.89 | 57.21 | 58.01 | 58.95 | 43.17 | 43.82 | 44.45 | 45.00 | 30.70 | 31.24 | 31.97 | 32.63 |
| JMLH | 46.58 | 48.15 | 48.74 | 48.89 | 57.53 | 58.55 | 59.13 | 60.94 | 44.47 | 45.56 | 45.92 | 46.02 | 31.38 | 32.64 | 33.19 | 33.77 |
| DPN | 44.57 | 47.08 | 48.09 | 48.56 | 56.82 | 57.43 | 58.26 | 59.83 | 44.87 | 45.72 | 46.11 | 46.19 | 31.11 | 31.65 | 31.59 | 32.13 |
| WGLHH | 47.92 | 49.83 | 50.19 | 52.35 | 57.17 | 57.99 | 58.46 | 59.23 | 45.11 | 45.91 | 46.68 | 47.32 | 32.21 | 33.13 | 33.74 | 34.11 |
| CSQ | 50.08 | 51.75 | 54.21 | 55.39 | 57.38 | 58.15 | 58.88 | 59.12 | 46.01 | 46.55 | 47.08 | 47.65 | 31.97 | 32.45 | 33.71 | 34.60 |
| OrH | 49.87 | 50.65 | 51.98 | 53.26 | 57.16 | 58.73 | 59.12 | 60.04 | 46.58 | 47.13 | 47.86 | 49.24 | 31.76 | 32.74 | 33.06 | 33.89 |
| REL | 50.39 | 51.21 | 52.64 | 53.97 | 58.68 | 59.02 | 59.86 | 60.35 | 47.05 | 47.67 | 48.18 | 48.86 | 32.47 | 33.27 | 34.29 | 35.01 |
| Jo-SRC | 50.82 | 51.42 | 51.96 | 53.62 | 58.12 | 58.91 | 59.89 | 60.87 | 47.79 | 48.14 | 48.97 | 49.34 | 32.59 | 33.19 | 34.61 | 35.11 |
| DIOR | 58.76 | 59.30 | 59.82 | 60.99 | 63.83 | 64.05 | 64.97 | 65.40 | 52.26 | 52.78 | 53.61 | 54.06 | 35.13 | 36.33 | 37.01 | 38.22 |
| STAR | 64.85 | 65.03 | 65.52 | 66.76 | 69.57 | 70.11 | 70.84 | 71.53 | 57.24 | 57.83 | 58.64 | 59.89 | 40.80 | 41.22 | 41.87 | 42.79 |
| **SEGA (Ours)** | **65.34** | **65.87** | **67.11** | **69.01** | **70.23** | **71.50** | **72.66** | **75.04** | **59.06** | **60.72** | **63.91** | **64.12** | **43.31** | **45.01** | **46.50** | **47.31** |

(a) Flickr25K (pair)   (b) Flickr25K (sym)   (c) NUS-WIDE (pair)   (d) NUS-WIDE (sym)

Figure 2: Performance comparison under different noise levels on Flickr25K and NUS-WIDE.

methods to ensure a fair comparison. Specifically, the first seven convolutional layers are fine-tuned, while the final fully connected hashing layer is trained from scratch. More details about the datasets and implementation can be found in the Appendix.

## 4.2 Empirical Results

**Performance Comparison.** We report the retrieval performance of all methods on four benchmark datasets under two types of label noise in Tables 1 and 2. Noise rates of both types are 60%. The results are measured by mean average precision (MAP) across different code lengths (16, 32, 64, and 128 bits). Table 1 shows the MAP scores under pairwise label noise. Table 2 reports the results under symmetric noise. Several clear observations can be made from the results. (1) Overall, SEGA achieves the highest MAP scores across all datasets and noise types, showing strong robustness to both structured and unstructured label corruption. SEGA clearly outperforms both conventional deep hashing models and recent noise-robust approaches. Specifically, SEGA achieves 64.61 MAP on NUS-WIDE with pairflip noise at 128 bits, more than 10% above DIOR and STAR. These findings confirm that SEGA can preserve semantic similarity under supervision noise. (2) Our SEGA consistently outperforms all the baselines, and as the dimension of the hash codes increases, the performance gap widens generally. This shows the strong scalability and robustness of our method in terms of the hash encoding dimension. (3) On MS COCO, which has dense and noisy multi-label annotations, our SEGA delivers particularly solid performance. This demonstrates its adaptability for real web data where the quality of annotations is poor.

**Effects of Different Noisy Rates.** Figure 2 shows the robustness of SEGA under different noise levels with 64-bit hash codes. Both pairwise and symmetric noise types are evaluated. As the noise rate increases from 0.2 to 0.8, most baseline methods drop sharply in performance. CSQ, Jo-SRC, and DPN degrade quickly beyond 0.4 noise rate, showing poor tolerance to heavy label corruption. In contrast, SEGA keeps consistently high MAP scores across all noise levels. The drop is smooth even when the noise rate reaches 0.8. For example, under pairflip noise on NUS-WIDE (Figure 2(c)), SEGA maintains a MAP above 60. It outperforms STAR and DIOR by large margins as noise increases. The advantage becomes most obvious at high noise levels, where other methods collapse but our structure-aware model stays stable. These results show that learning semantic geometry and calibrating supervision jointly gives better robustness than label correction alone.

**Ablation Study.** We evaluate the contribution of each module in SEGA through ablation experiments, where one loss term is removed at a time. Table 3 reports the MAP scores under symmetric and pairflip noise on four datasets. Removing $\mathcal{L}_{\text{clean}}$ results in a clear drop across all datasets, showing that prototype-guided supervision is essential for stable learning. When $\mathcal{L}_{\text{cont}}$ is excluded, performance

Table 3: Ablation studies on four benchmarks under symmetric and pairflip label noise.

| Method | CIFAR-10 | | | | Flickr25K | | | | NUS-WIDE | | | | MS COCO | | | |
|---|---|---|---|---|---|---|---|---|---|---|---|---|---|---|---|---|
| Noise Type.Bits | sym.32 | pair.32 | sym.64 | pair.64 | sym.32 | pair.32 | sym.64 | pair.64 | sym.32 | pair.32 | sym.64 | pair.64 | sym.32 | pair.32 | sym.64 | pair.64 |
| SEGA w/o $\mathcal{L}_{clean}$ | 60.61 | 68.63 | 61.62 | 72.03 | 62.04 | 70.02 | 62.18 | 67.81 | 51.52 | 54.75 | 51.88 | 58.70 | 41.37 | 41.41 | 41.55 | 40.22 |
| SEGA w/o $\mathcal{L}_{cont}$ | 64.14 | 69.86 | 64.59 | 71.20 | 70.25 | 71.24 | 70.40 | 74.26 | 60.03 | 56.31 | 61.28 | 58.05 | 43.12 | 44.36 | 45.11 | 47.06 |
| SEGA w/o $\mathcal{L}_{calib}$ | 62.95 | 70.45 | 66.33 | 71.87 | 68.47 | 70.10 | 68.37 | 73.35 | 59.09 | 58.05 | 62.70 | 59.34 | 44.89 | 44.85 | 42.23 | 43.74 |
| SEGA w/o $\mathcal{L}_{mix}$ | 60.48 | 70.12 | 63.05 | 71.39 | 68.26 | 69.94 | 68.46 | 73.38 | 58.72 | 57.34 | 61.65 | 60.70 | 44.20 | 45.23 | 44.84 | 46.90 |
| **SEGA (Ours)** | **65.87** | **70.59** | **67.11** | **72.34** | **71.50** | **72.14** | **72.66** | **74.49** | **60.72** | **59.52** | **63.91** | **61.86** | **45.01** | **46.08** | **46.50** | **47.21** |

(a) Flickr25K    (b) NUS-WIDE    (c) Baseline    (d) Ours

Figure 3: (a)(b) Sensitivity analysis of percentile threshold $q_r$ with 64-bit hash codes. (c)(d) t-SNE visualization of 128-bit hash codes on CIFAR-10 under symmetric noise. The baseline for comparison is STAR, the strongest competing method.

decreases, especially under pairflip noise, indicating its necessity to maintain structural consistency on noisy areas. Excluding $\mathcal{L}_{calib}$ compromises robustness to both settings of noise and confirms that soft supervision limits the impact of annotation errors. Removing $\mathcal{L}_{mix}$ also causes consistent performance loss, particularly on MS COCO, demonstrating the advantage of boundary regularization. Collectively, these results demonstrate the necessity of the elements of SEGA and their effectiveness in complementing each other. The Appendix contains additional ablation results.

**Sensitivity analysis.** We evaluate the sensitivity of SEGA to hyperparameter $q_r$ under pairflip and symmetric noise with 64-bit hash codes. The best performance appears when the $q_r$ value is set to 0.3 or 0.5, from Figure 3(a) and 3(b). Performance drops when $q_r$ is too high, as excessive filtering removes many clean samples. Denoising becomes weak when $q_r$ is too low, since noisy samples remain unfiltered. These observations imply a moderate threshold to achieve the best compromise between over-filtering and under-filtering.

**Visualization.** To analyze the learned semantic structure, we visualize the hash embeddings using t-SNE on the CIFAR-10 dataset. The experiment is performed under symmetric noise with 128-bit hash codes. Specifically, SEGA generates clusters that are far more separated than STAR, as shown in Figure3(c) and 3(d). Our SEGA presents a clearer boundary for each category, which indicates our framework learns more discriminative hash embeddings. Conversely, STAR shows evident overlaps between classes, demonstrating lower robustness to label noise. These results show SEGA learns hash codes that are semantically consistent and robust against noisy supervision.

# 5 Conclusion

In this paper, we present a unified framework named SEGA for robust multi-label hashing under noisy supervision. The method models semantic geometry to improve the reliability of learned hash representations. We first use prototype-guided semantic anchoring to align embeddings with class semantics and maintain structural stability. Next, we design an uncertainty-guided calibration module to adjust the effect of unreliable labels. We also apply a structure-aware interpolation strategy to smooth decision boundaries and enhance local consistency in noisy regions. In theory, SEGA reduces semantic misalignment and improves class boundaries. Experiments on several noisy hashing benchmarks show clear robustness gains. Although built for multi-label hashing, the core ideas of SEGA can be extended to other structure-sensitive learning tasks, such as multi-modal retrieval and multi-label classification under weak or noisy supervision.

# Acknowledgement

Ming Zhang and Yiyang Gu are supported by grants from the National Key Research and Development Program of China with Grant No. 2023YFC3341203 and the National Natural Science Foundation of China (NSFC Grant Number 62276002).

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

## A  More Experiment Results

We conduct ablation studies on 16-bit and 128-bit hash codes. As shown in Table 4, removing any loss term leads to a clear drop in performance across all the datasets. This demonstrates that each module plays a necessary role in learning robust hash codes against label noise. For instance, removing $\mathcal{L}_{\text{calib}}$ causes clear drops in performance across all the datasets, especially with 16-bit hash codes. This justifies the benefit of uncertainty-guided supervision calibration in noisy environments. Turning off $\mathcal{L}_{\text{mix}}$ also reduces MAP, particularly on Flickr25K, showing its effect in regularizing category boundaries. These results confirm that all modules in our framework are complementary and necessary under different code lengths.

Table 4: Ablation studies on four benchmarks under symmetric (sym.) and pairflip (pair.) label noise at 16-bit and 128-bit code lengths.

| Method | cifar-10 | | | | Flickr25K | | | | NUS-WIDE | | | | MS-COCO | | | |
|---|---|---|---|---|---|---|---|---|---|---|---|---|---|---|---|---|
| Noise Type.Bits | sym.16 | pair.16 | sym.128 | pair.128 | sym.16 | pair.16 | sym.128 | pair.128 | sym.16 | pair.16 | sym.128 | pair.128 | sym.16 | pair.16 | sym.128 | pair.128 |
| SEGA w/o $\mathcal{L}_{\text{clean}}$ | 59.70 | 64.87 | 62.85 | 70.96 | 60.71 | 63.92 | 63.72 | 68.82 | 52.54 | 54.74 | 55.17 | 62.87 | 37.85 | 38.03 | 43.80 | 44.24 |
| SEGA w/o $\mathcal{L}_{\text{cont}}$ | 62.98 | 63.49 | 65.90 | 72.96 | 69.38 | 69.78 | 71.85 | 75.13 | 57.66 | 57.07 | 62.18 | 60.68 | 42.75 | 42.38 | 46.26 | 45.33 |
| SEGA w/o $\mathcal{L}_{\text{calib}}$ | 62.37 | 67.30 | 64.34 | 73.07 | 66.08 | 68.19 | 70.16 | 75.74 | 54.19 | 56.40 | 63.20 | 63.12 | 41.50 | 43.83 | 46.38 | 46.88 |
| SEGA w/o $\mathcal{L}_{\text{mix}}$ | 61.93 | 67.94 | 68.66 | 73.38 | 66.12 | 68.08 | 69.80 | 75.71 | 53.66 | 58.46 | 61.65 | 64.43 | 42.81 | 41.98 | 45.94 | 45.57 |
| **SEGA (Ours)** | **65.34** | **70.01** | **69.01** | **73.56** | **70.23** | **71.19** | **75.04** | **76.48** | **59.06** | **59.11** | **64.12** | **64.61** | **43.31** | **44.37** | **47.31** | **47.58** |

## B  Proofs of Theorems

**Theorem 3.1** (Uncertainty-Weighted Confidence Approximates Label Correctness). *Let $x_i$ be a training sample with observed label $\tilde{y}_i$ and true (latent) label $y_i^*$. Define the energy-based prediction confidence as $E_i = -\log \sum_{c=1}^{C} \exp(l_{ic})$, and $\tilde{E}_i = \frac{E_i - \min_j E_j}{\max_j E_j - \min_j E_j}$, the structure-base divergence as $\delta_i = 1 - \frac{1}{K} \sum_{j \in \mathcal{N}_i} \cos(\hat{h}_i, \hat{h}_j)$. If we assume the probability that the label is correct depends solely on the semantic and structural reliability of the sample, and these two sources are conditionally independent given $x_i$, then:*

$$\Pr[y_i^* = \tilde{y}_i \mid x_i] \propto 1 - U_i. \tag{16}$$

*Proof.* The probability that a sample is confidently predicted is proportional to the negative energy in energy-based learning [40]: $\Pr[\text{semantic } \text{correctness}|x_i] \propto \exp(-E_i)$.

After batch normalization, $\tilde{E}_i \in [0,1]$ maintains a monotonic relationship with $E_i$ and thus $\Pr[\text{semantic } \text{correctness}|x_i] \propto 1 - \tilde{E}_i$.

The structural divergence score $\delta_i$ reflects how well a sample aligns with its local neighborhood. Samples in high-density regions with consistent semantic structure are more likely to have correct labels under the widely adopted cluster assumption. In contrast, geometrically isolated samples, those with low average similarity to neighbors, are often located near semantic boundaries or in underrepresented regions of the data manifold and are thus more prone to label noise.

This assumption is supported by several foundational approaches. In graph-based semi-supervised learning [70, 3], Laplacian regularization encourages neighboring nodes to share similar outputs. Label propagation methods [8] assume that labels should remain consistent within locally coherent regions. Moreover, the low-density separation principle [16] suggests that well-formed clusters of correctly labeled data tend to lie in structurally connected, high-confidence zones. Therefore, a smaller divergence score (i.e., a larger $1 - \delta_i$) indicates stronger neighborhood support and a higher likelihood of label correctness. We thus approximate structural confidence as being proportional to the complement of divergence:

$$\Pr[\text{structural } \text{consistency}|x_i] \propto 1 - \delta_i.$$

According to the assumption, we can obtain

$$\Pr[y_i^* = \tilde{y}_i|x_i] \propto (1 - \tilde{E}_i)(1 - \delta_i).$$

On the other hand,

$$1 - U_i = 1 - (1 - \tilde{E}_i)\delta_i = (1 - \tilde{E}_i)(1 - \delta_i) - \tilde{E}_i\delta_i,$$

when $\tilde{E}_i$ and $\delta_i$ are relatively small (i.e., low uncertainty), the second term is small, and
$$1 - U_i \approx (1 - \tilde{E}_i)(1 - \delta_i) \propto \Pr[\boldsymbol{y_i^*} = \tilde{\boldsymbol{y}}_i | \boldsymbol{x_i}].$$

Under the assumption that label correctness depends only on semantic and structural confidence, modeled respectively by energy-based and neighborhood-based signals, the complement of the joint uncertainty score $1 - U_i$ provides a proportional estimate of the posterior label correctness probability. $\qquad\square$

**Theorem 3.2** (Structure-Preserving Interpolation Bound). *Let $\boldsymbol{x}_i$ and $\boldsymbol{x}_j$ be two clean samples belonging to class c, i.e., $\boldsymbol{y}_{ic} = \boldsymbol{y}_{jc} = 1$. $\hat{\boldsymbol{h}}_i$ and $\hat{\boldsymbol{h}}_j$ represent the normalized hash embeddings of these two samples. Let $\boldsymbol{p}_c$ be the unit-norm prototype vector of class c. For any $\lambda \in [0,1]$, the interpolated embedding is defined as follows:*
$$\tilde{\boldsymbol{h}} = \lambda \hat{\boldsymbol{h}}_i + (1 - \lambda)\hat{\boldsymbol{h}}_j.$$
*The cosine similarity between $\tilde{\boldsymbol{h}}$ and $\boldsymbol{p}_c$ has the following lower bound:*
$$\cos(\tilde{\boldsymbol{h}}, \boldsymbol{p}_c) \geq \lambda \cos(\hat{\boldsymbol{h}}_i, \boldsymbol{p}_c) + (1 - \lambda)\cos(\hat{\boldsymbol{h}}_j, \boldsymbol{p}_c).$$

*Proof.* Since both $\hat{\boldsymbol{h}}_i$ and $\hat{\boldsymbol{h}}_j$ are normalized to unit norm (i.e., $\|\hat{\boldsymbol{h}}_i\| = \|\hat{\boldsymbol{h}}_j\| = 1$), and $\boldsymbol{p}_y$ is also a unit-norm vector (i.e., $\|\boldsymbol{p}_y\| = 1$), we begin by expanding the cosine similarity between $\tilde{\boldsymbol{h}}$ and $\boldsymbol{p}_y$:
$$\cos(\tilde{\boldsymbol{h}}, \boldsymbol{p}_c) = \frac{\tilde{\boldsymbol{h}}^\top \boldsymbol{p}_c}{\|\tilde{\boldsymbol{h}}\|} = \frac{\left(\lambda \hat{\boldsymbol{h}}_i + (1 - \lambda)\hat{\boldsymbol{h}}_j\right)^\top \boldsymbol{p}_c}{\|\tilde{\boldsymbol{h}}\|}.$$
Applying linearity of inner product:
$$\tilde{\boldsymbol{h}}^\top \boldsymbol{p}_c = \lambda \hat{\boldsymbol{h}}_i^\top \boldsymbol{p}_c + (1 - \lambda)\hat{\boldsymbol{h}}_j^\top \boldsymbol{p}_c = \lambda \cos(\hat{\boldsymbol{h}}_i, \boldsymbol{p}_c) + (1 - \lambda)\cos(\hat{\boldsymbol{h}}_j, \boldsymbol{p}_c).$$

Thus,
$$\cos(\tilde{\boldsymbol{h}}, \boldsymbol{p}_c) = \frac{\lambda \cos(\hat{\boldsymbol{h}}_i, \boldsymbol{p}_c) + (1 - \lambda)\cos(\hat{\boldsymbol{h}}_j, \boldsymbol{p}_c)}{\|\tilde{\boldsymbol{h}}\|}.$$

Since $\|\tilde{\boldsymbol{h}}\| \leq 1$ by convexity of Euclidean norm (specifically, Minkowski inequality), we conclude:
$$\cos(\tilde{\boldsymbol{h}}, \boldsymbol{p}_c) \geq \lambda \cos(\hat{\boldsymbol{h}}_i, \boldsymbol{p}_c) + (1 - \lambda)\cos(\hat{\boldsymbol{h}}_j, \boldsymbol{p}_c).$$
as desired. $\qquad\square$

## C  Dataset Details

We evaluate our method on four widely used image retrieval benchmarks. All datasets are standard in deep hashing and image retrieval literature, with consistent splits across prior work [55].

**CIFAR-10** [28, 30]. This is a single-label dataset containing 60,000 natural images evenly divided into 10 categories. Each image is $32 \times 32$ in resolution with three color channels. We sample 1,000 examples per class as queries. The remaining examples are utilized as the retrieval database, from which 500 images per class are further sampled for training. The low resolution and compact structure make this dataset a challenging testbed for semantic feature extraction.

**Flickr25K** [24]. The Flickr25K dataset collects 25,000 multi-label images from the Flickr platform. Each image in the dataset is tagged with one or more labels from 38 semantic concepts. We utilize the 24 most frequent concepts as the label set following [55]. We use 2,000 images as queries, and sample 10,000 images for training from the remaining retrieval set. The annotations are sparse in this dataset, where each image is assigned 4.7 labels on average.

**NUS-WIDE** [10]. The NUS-WIDE dataset includes 269,648 web images with 81 semantic tags. We utilize the 10 most frequent tags as the target categories following [55]. We employ 5,000 images as queries, and sample 5,000 images from the remaining retrieval set for training.

**MS COCO** [36, 58]. The MS COCO dataset contains more than 120,000 images, where each image is tagged by some of 80 categories. We adopt the 2014 release and use 5,000 images as queries. From the remaining retrieval set, 10,000 are sampled for training. Its complex scenes and dense annotations make it a challenging dataset for fine-grained image retrieval.

## D Implementation Details

All experiments are implemented in PyTorch and run on a single NVIDIA A40 GPU in a standard Linux environment. Two types of label noise are introduced. In symmetric noise, each label is randomly replaced by another class with equal probability. In pairwise flip noise, labels are switched to semantically related categories according to predefined mappings [55]. The noise rate ranges from 20% to 80% in increments of 20%. During mixup training, interpolation coefficients are sampled from a symmetric Beta distribution with $\alpha = 0.4$ as in [66]. The percentile threshold $q_r$ for selecting clean samples is set to 0.3 by default. The code can be found at `https://github.com/dllab001/SEGA`.

## E Broader Impacts and Limitations

This paper deals with a problem of robust image retrieval under noisy supervision and proposes a generic framework called SEGA to do so. With this process our approach is scalable and reliable in real-world settings where label quality is imperfect: e.g. web-based image indexing and large-scale multimedia systems. As well, although our framework is instantiated in multi-label hashing, its essential principles are not domain-specific and can be readily adapted to more general tasks, e.g. multi-modal retrieval, multi-label classification and other structure-sensitive learning tasks under weak or noisy supervision. However, one limitation of our current solution is that we use a static retrieval database, which does not have the ability to cope with the change in the data distribution in dynamic environments. Future work could adapt SEGA to continual learning or online retrieval situations.

