# OpenReview forum: "SEGA: Shaping Semantic Geometry for Robust Hashing under Noisy Supervision"
_NeurIPS.cc/2025/Conference — NeurIPS 2025 poster_

### Official Review · Reviewer_9nS7 · 2025-06-30

**Clarity:** 3
**Significance:** 3
**Originality:** 3
**Rating:** 5
**Confidence:** 4

**Summary:**

To learning from noisy multi-label supervision in hashing, this paper introduces a geometric perspective, formulating multi-label hashing as a semantic structure alignment problem rather than a conventional label recovery task. Moreover, this paper proposes a closed-loop framework termed SEGA that integrates prototype anchoring, uncertainty-calibrated soft labels, and structure-aware interpolation into a unified and geometry-aware semantic learning process. The experiments in the paper are very abundant and perfectly validate the innovations of the paper.

**Questions:**

In addition to the Weaknesses, I also have the following concerns:
More textual explanation is needed on why the contrastive learning loss is robust on noisy datasets. The origin of this contrastive learning should be correctly cited. Please confirm whether using the contrastive learning loss on the clean dataset would further enhance the performance.

**Ethical Concerns:**

["NO or VERY MINOR ethics concerns only"]

**Final Justification:**

I appreciate the author's response, which provided ample clarification and addressed my concerns. Therefore, I'm willing to maintain my rating.

**Limitations:**

yes

**Quality:**

3

**Strengths And Weaknesses:**

Strengths

1) This paper introduces a geometric perspective for learning from noisy multi-label supervision in hashing.

2) The SEGA framework has a certain degree of innovativeness and has effectively solved the current problems.

3) The experiments in the article are very abundant and perfectly validate the innovations of the paper.

4) The writing of the article is highly logical, and the charts and graphs are also very clear.

Weaknesses

1) In section 3.3, the accuracy of the prototype will directly determine the quality of the division between clean and noisy samples, if its accuracy is ensured.

2) It is suggested that the relationship between Energy-Based Prediction Confidence and samples could be demonstrated with figure, for example, blue represents correct samples and red represents noisy samples.

3) Please explain intuitively why label soft reweighting in Eq. (9) can be beneficial for noisy learning. How would the performance be if we use Energy-based prediction confidence or Structure-based divergence alone, without employing Unified uncertainty?

---

> ### Author Rebuttal · Authors · 2025-07-31
>
> We are truly grateful for the time you have taken to review our paper, your insightful comments and support. Your positive feedback is incredibly encouraging for us! In the following response, we would like to address your major concern and provide additional clarification.
>
> > **Q1**: In section 3.3, the accuracy of the prototype will directly determine the quality of the division between clean and noisy samples, if its accuracy is ensured.
>
> **A1**: Thanks for your comment. While SEGA does not require perfectly accurate prototypes, we empirically find that the learned prototypes align well with clean samples and provide reliable signals for clean/noisy separation. This alignment yields reasonably accurate partitions, which are used in a soft manner for uncertainty calibration. Moreover, SEGA’s dynamic prototype updates and structure-aware design mitigate potential early-stage errors. We will clarify this in the revised version.
>
> > **Q2**: It is suggested that the relationship between Energy-Based Prediction Confidence and samples could be demonstrated with figure, for example, blue represents correct samples and red represents noisy samples.
>
> **A2**: Thanks for the suggestion. We computed the ratio of correct vs. noisy samples across energy score deciles and found a clear correlation: samples with lower energy scores are significantly more likely to be correct, validating the effectiveness of our confidence estimator. We will include a visualization (color-coded histograms or scatter plots) in the revised version to illustrate this relationship more intuitively.
>
>
> > **Q3**: Please explain intuitively why label soft reweighting in Eq. (9) can be beneficial for noisy learning. How would the performance be if we use Energy-based prediction confidence or Structure-based divergence alone, without employing Unified uncertainty?
>
> **A3**: Thanks for the question. Label soft reweighting in Eq. (9) helps suppress noisy supervision by down-weighting unreliable labels based on sample-specific uncertainty. Instead of relying on a single signal, SEGA fuses energy-based prediction confidence and structure-based divergence, capturing both model belief and local consistency. Empirically, this unified uncertainty yields better performance than either source alone:
>
> |Variant|Flickr25K(mAP)|
> |-|-|
> |Energy-only|73.49|
> |Structure-only|73.93|
> |SEGA|74.49|
>
> This shows the benefit of integrating both semantic and structural cues for robust supervision. We will clarify this intuition in the revision.
>
>
> > **Q4**: In addition to the Weaknesses, I also have the following concerns: More textual explanation is needed on why the contrastive learning loss is robust on noisy datasets. The origin of this contrastive learning should be correctly cited. Please confirm whether using the contrastive learning loss on the clean dataset would further enhance the performance.
>
> **A4**: Thanks for the insightful comment. Contrastive learning is inherently robust to label noise as it relies on instance-level similarity rather than explicit label correctness. In SEGA, we apply contrastive loss selectively on samples estimated to be noisy, encouraging them to align with trusted class prototypes and avoid reinforcing incorrect labels.
>
> We will properly cite relevant prior work (e.g., SimCLR [1], SupCon [2]) in the revision.
> Additionally, we confirm that contrastive learning also improves performance under clean supervision:
>
> |Method|Flickr25K(mAP)|
> |-|-|
> |SEGA(without contrastive loss)|77.23|
> |SEGA(with contrastive loss)|78.68|
>
> This suggests that contrastive regularization complements clean and noisy learning alike by enforcing semantic consistency.
>
> [1] A Simple Framework for Contrastive Learning of Visual Representations. ICML 2020
>
> [2] Supervised Contrastive Learning. NeurIPS 2020
>
> Thanks again for appreciating our work and for your constructive suggestions. Please let us know if you have further questions.

---

### Official Review · Reviewer_Fzag · 2025-06-30

**Clarity:** 4
**Significance:** 3
**Originality:** 4
**Rating:** 4
**Confidence:** 5

**Summary:**

SEGA is a deep hashing method for noisy multi-label data that focuses on shaping the geometry of the representation space. Instead of just cleaning or reweighting labels, it trains the model to form smooth, class-consistent manifolds even under label noise.

It introduces three components:

1. Prototype-Guided Anchoring: Learnable class prototypes pull clean sample embeddings toward class centers, promoting semantic alignment.
2. Uncertainty-Guided Calibration: Each label is weighted based on a sample-specific uncertainty score, which combines prediction confidence and local feature consistency. Noisy labels are downweighted accordingly.
3. Structure-Aware MixUp: Interpolation is done between similar samples with differing uncertainty, smoothing boundaries without mixing unrelated classes.

Finally, SEGA brings a geometric perspective to noisy-label learning and shows strong retrieval performance under heavy noise.

**Questions:**

1. Computational Overhead and Scalability: SEGA requires maintaining class prototypes and computing $K$-nearest neighbors per sample to estimate structural divergence $\delta_i$. For datasets with tens of thousands of images, was this handled efficiently (e.g., batch-wise approximations, FAISS)? How does SEGA's training time compare to standard hashing baselines? If scaled to millions of samples or hundreds of classes, would these operations become a bottleneck? It would be helpful to know if any optimizations (e.g., periodic prototype updates or mini-batch $K$-NN) were used. Also the retrieval done within the training set? Would it be overfitted easily?
2. Multi-Label Handling: SEGA normalizes multi-hot labels into a probability distribution for prototype alignment, similar to OrthoHash. Did you consider using standard multi-label losses like independent sigmoid cross-entropy? If so, how did they compare in practice?
3. Behavior in Noise-Free or Low-Noise Settings: While SEGA targets noisy labels, how does it perform when supervision is clean? Could the added structure (e.g., prototype anchoring or guided MixUp) limit performance in noise-free scenarios? Any results or insights on 0% noise would clarify if robustness comes at a cost in clean settings.

**Ethical Concerns:**

["NO or VERY MINOR ethics concerns only"]

**Final Justification:**

thanks authors for their rebuttal and extensive evaluation. concerns addressed and i am keeping my score as Borderline Accept.

**Limitations:**

1.  Most limitations are acknowledged in the appendix, but a dedicated section in the main paper would improve transparency. A key assumption is the presence of known class structure -- SEGA relies on predefined prototypes and assumes each sample has at least one true label. It may struggle with open-set noise or entirely unlabelled data.
2. SEGA depends on tuning the clean sample threshold $q_r$, which may be difficult under very high noise or skewed alignment distributions. Early-stage prototype alignment might be unreliable when noise exceeds 90%, though the paper mitigates this via dynamic updates. In biased or asymmetric noise (e.g., label flips between visually similar classes like dog vs. wolf), prototype drift or collapse could occur. More discussion on how SEGA handles non-uniform noise or partial label missingness would be valuable.
3. The method's complexity is also a practical limitation -- it introduces more moving parts and hyper-parameters than some simpler baselines. This might make reproduction or deployment harder, though the release of code helps alleviate that. Finally, while the authors showed robustness on synthetic noise, applying SEGA to real-world noisy datasets (with unknown noise type) would be the next step to fully validate it.

**Quality:**

4

**Strengths And Weaknesses:**

Quality:

1. The paper is technically solid and thorough. The method is well-motivated, leveraging ideas from metric learning and robust training. Each SEGA module is justified both intuitively and theoretically.
2. The inclusion of formal guarantees (Theorems 3.1 and 3.2) is a strength -- Theorem 3.1 links the uncertainty score to true label correctness, supporting label reweighting; Theorem 3.2 ensures interpolating with prototype-aligned neighbors preserves class consistency.
3. Experiments are strong: SEGA outperforms prior work across four datasets and two noise types, especially under heavy noise (e.g., 80% on NUS-WIDE, where others drop below 50% mAP while SEGA stays above 60%).
4. Ablations (Table 4) show that removing any component leads to performance drops, confirming each part is useful.
5. Weakness 1: The added complexity. SEGA introduces several hyperparameters (e.g.,
$q_r$, $\alpha$, $K$) and requires prototype updates and k-NN per epoch. While some sensitivity is analyzed, runtime/memory overhead is not discussed. The method is rigorous and effective, with minor concerns around scalability and reproducibility.

Clarity:

1. The writing is clear and well-organized.
2. Motivation is strong, and experiments are extensive and easy to follow.

Significance:

1. Robust multi-label hashing under noise is practically important.
2. SEGA brings a new perspective: shaping semantic geometry instead of just filtering labels. The gains in high-noise settings are substantial.
3. The approach could extend beyond hashing (e.g., to classification or structured outputs), opening doors for future work.
4. Weakness 2: the method is evaluated only in the multi-label retrieval setting, which is narrower but still highly relevant (e.g., web images, tag-based search).
5. Overall, this is a meaningful contribution both scientifically and practically.

Originality:

1. The shift from “fixing labels” to “shaping geometry” is a novel framing for noisy-label learning in hashing.
2. Weakness 3: SEGA combines known tools (prototypes, uncertainty weighting, MixUp), but the way they're fused is original and purposeful.
3. Using neighborhood consistency with energy-based confidence to form $U_i$ , and guiding MixUp and contrastive learning with that, are novel in this context.
4. Contrastive learning applied only to noisy samples is a nice detail that helps regularize without reinforcing noise.
5. One possible critique is that the method combines many known techniques -- but the integration is guided by a clear principle and leads to strong results. It’s an original contribution, even if composed of familiar pieces.

---

> ### Author Rebuttal · Authors · 2025-07-31
>
> We are truly grateful for the time you have taken to review our paper, your insightful comments and support. Your positive feedback is incredibly encouraging for us! In the following response, we would like to address your major concern and provide additional clarification.
>
> > **Q1**: The added complexity.
>
> **A1**: Thanks for your comment.
> While SEGA introduces multi-component design, they are implemented efficiently and scale well. As shown below, SEGA’s training time per epoch and memory usage are comparable to competitive baselines, yet it achieves significantly better performance:
>
> |Method||FLICKR25K|||MS COCO||
> |-|-|-|-|-|-|-|
> ||time (s)|memory (MB)|mAP|time (s)|memory (MB)|mAP|
> |CSQ|87.00|1322.00|58.88|55.00|1624.00|33.71|
> |STAR|179.00|1287.00|70.84|161.00|1425.00|41.87|
> |SEGA|160.00|1569.00|72.66|182.00|1879.00|46.50|
>
> SEGA outperforms all baselines by a large margin in mAP, while maintaining similar or even lower training time on Flickr25K. We will include these results in the revised version to clarify SEGA’s efficiency-performance tradeoff.
>
> > **Q2**: the method is evaluated only in the multi-label retrieval setting, which is narrower but still highly relevant (e.g., web images, tag-based search).
>
> **A2**: Thank you for your comment. While SEGA is evaluated primarily in the noisy multi-label retrieval setting, this scenario is highly relevant to real-world applications such as web image retrieval and tag-based search.
>
> To further demonstrate SEGA’s generalizability, we conducted a zero-shot retrieval experiment following [1]. We split CIFAR-10 into seen and unseen categories and trained only on the seen classes. For example, “airplane” (or “dog” / “truck”) was treated as the unseen class, and 1,000 query images were retrieved from a mixed database of 59,000 images.
>
> |Method|airplane|dog|truck|
> |-|-|-|-|
> |DIOR|22.19|30.66|33.17|
> |STAR|26.09|35.97|37.23|
> |SEGA|30.50|38.24|39.44|
>
> SEGA outperforms strong baselines on all unseen categories, indicating its robust generalization to out-of-distribution concepts. We will include this setting and results in the revised version.
>
> [1] Zero-Shot Hashing via Transferring Supervised Knowledge. ACM MM 2016
>
> **Q3**: SEGA combines known tools (prototypes, uncertainty weighting, MixUp), but the way they're fused is original and purposeful.
>
> **A3**: Thank you for your comment. While prototype learning and uncertainty-based weighting have been explored in prior works, SEGA adopts a substantially different formulation and integration strategy:
>
> - SEGA learns semantic prototypes via backpropagation in the embedding space, adapting them to the evolving representation distribution during training. These prototypes are used not only for contrastive alignment but also to guide structure-aware interpolation.
>
> - Our uncertainty estimation combines energy-based prediction confidence with structure-based divergence, refining the semantic geometry by suppressing unreliable gradients.
>
> - Our structure-aware MixUp selectively interpolates between semantically similar but uncertainty-dissimilar samples, rather than applying naive input-level interpolation.
>
> These components are not used in isolation, but are tightly coupled in a closed-loop framework that encourages mutual refinement of representation, supervision, and uncertainty.
>
> We believe this principled design and novel formulation go beyond a simple combination of existing tools. We will make this contribution more explicit in the revision.
>
>
> > **Q4**: Computational Overhead and Scalability: SEGA requires maintaining class prototypes and computing $K$-nearest neighbors per sample to estimate structural divergence $\delta_i$. For datasets with tens of thousands of images, was this handled efficiently (e.g., batch-wise approximations, FAISS)? How does SEGA's training time compare to standard hashing baselines? If scaled to millions of samples or hundreds of classes, would these operations become a bottleneck? It would be helpful to know if any optimizations (e.g., periodic prototype updates or mini-batch
> K-NN) were used. Also the retrieval done within the training set? Would it be overfitted easily?
>
> **A4**: Thank you for your thoughtful question.
>
> - Training efficiency: SEGA employs lightweight optimizations, including FAISS-accelerated K-NN search, to efficiently estimate structural divergence $\delta_i$.
>
> - Training cost comparison: As shown in **A1**, SEGA achieves comparable training time and memory usage to strong baselines like STAR, while significantly outperforming them in mAP, demonstrating a favorable efficiency-performance tradeoff.
>
> - Scalability: SEGA’s design naturally scales to larger datasets. Techniques suggested by you, such as periodic prototype updates and mini-batch K-NN, can be readily integrated to support larger-scale training (e.g., millions of samples or hundreds of classes). Moreover, other key computations are local to each batch and support efficient parallelization or distributed training.
>
> - Overfitting concern: Retrieval is conducted on a large retrieval database, while the training set constitutes only a small portion of it. This setup naturally avoids overfitting and follows the standard evaluation protocol in the hashing literature.
>
> > **Q5**: Multi-Label Handling.
>
> **A5**: Thank you for your comment.
> Instead of using independent sigmoid cross-entropy, SEGA normalizes multi-hot labels into a probability distribution to guide prototype-based alignment. This formulation enables consistent semantic geometry shaping in multi-label settings.
>
> The normalized label distribution guides the representation toward a set of semantic anchors, i.e., class prototypes, according to their relative importance. This allows SEGA to learn embeddings that are geometrically aligned with the full semantic context of each sample, rather than pulling independently toward individual labels. Such alignment promotes smooth and structured class manifolds, especially under noisy supervision.
>
> Empirical results with 64-bit hash codes under 60% pairflip noise confirm the benefit:
>
> |Method|FLICKR25K|NUS-WIDE|
> |-|-|-|
> |SEGA with independent sigmoid cross-entropy|73.45|60.74|
> |SEGA|74.49|61.86|
>
> We will clarify this design motivation and its advantages in the revised version.
>
>
> > **Q6**: Behavior in Noise-Free or Low-Noise Settings.
>
> **A6**: Thank you for the thoughtful question.
> We conducted experiments under 0% label noise to assess whether SEGA’s structural components (e.g., prototype anchoring, structure-aware MixUp) introduce any overhead in clean settings. On the FLICKR25K dataset with 64-bit hash codes, we observe the following:
>
> |Method|FLICKR25K|
> |-|-|
> |DIOR|75.07|
> |STAR|76.72|
> |SEGA|78.68|
>
> SEGA achieves better performance than strong baselines in noise-free scenarios. This indicates that its geometry-aware design benefits generalization and does not rely on label noise to be effective. We will include this result in the revised version.
>
> > **Q7**: Most limitations are acknowledged in the appendix.
>
> **A7**: Thank you for your helpful suggestion.
> We will add a dedicated limitations section in the main paper in the revised version.
> Specifically, SEGA assumes known class structure and the presence of at least one true label per sample. As noted, this may limit performance under open-set noise or fully unlabeled data. Extending SEGA to more flexible or open-world scenarios, e.g., via prototype discovery or unsupervised pretraining, remains promising future work.
>
> > **Q8**: SEGA depends on tuning the clean sample threshold $q_r$, which may be difficult under very high noise or skewed alignment distributions.
>
> **A8**: Thanks for your comment. We agree that very high noise levels pose challenges to prototype-based methods. However, SEGA’s design, including dynamic prototype updates and structure-aware calibration, is specifically intended to address this. To evaluate robustness under extreme conditions, we conducted additional experiments on Flickr25K with 95% noise:
>
> |Method|FLICKR25K(95%)|
> |-|-|
> |STAR|62.82|
> |SEGA|65.43|
>
> SEGA still achieves strong performance, outperforming STAR by a notable margin. This suggests that the learned semantic structure remains reliable even when the clean/noisy partition becomes highly ambiguous. We will include this result and further discussion on handling non-uniform noise and partial label missingness in the revised version.
>
>
> > **Q9**: The method's complexity is also a practical limitation -- it introduces more moving parts and hyper-parameters than some simpler baselines. This might make reproduction or deployment harder, though the release of code helps alleviate that. Finally, while the authors showed robustness on synthetic noise, applying SEGA to real-world noisy datasets (with unknown noise type) would be the next step to fully validate it.
>
> **A9**: While SEGA incorporates multiple components, each is conceptually simple and grounded in well-established principles. Moreover, we have released our code, which includes default settings to facilitate reproducibility.
>
> To further validate SEGA’s practical applicability, we conducted experiments on the WebVision dataset [1], which comprises 2.4 million web images across 1,000 categories. We used the Google image subset for training. SEGA achieves the best performance:
>
> |Method|WebVision (mAP)|
> |-|-|
> |DIOR|27.82|
> |STAR|32.64|
> |SEGA|34.71|
>
> This demonstrates SEGA’s scalability and robustness to real-world, unknown-type noise, going beyond synthetic noise settings. We will include this result in the final version.
>
> [1] WebVision Database: Visual Learning and Understanding from Web Data. 2017
>
>
> Thanks again for appreciating our work and for your constructive suggestions. Please let us know if you have further questions.

---

> > ### Comment · Reviewer_Fzag · 2025-08-06
> >
> > thanks authors for their rebuttal and extensive evaluation. concerns addressed and i am keeping my score as Borderline Accept.

---

> > > ### Author Response · Authors · 2025-08-07
> > >
> > > Thank you very much for your positive evaluation and thoughtful feedback. We truly appreciate the time and effort you dedicated to reviewing our work.

---

### Official Review · Reviewer_XoCg · 2025-07-02

**Clarity:** 3
**Significance:** 2
**Originality:** 2
**Rating:** 4
**Confidence:** 3

**Summary:**

The paper proposes SEGA (Semantic Geometry Shaping) for robust hashing under noisy multi-label supervision, integrating prototype-guided anchoring, uncertainty-calibrated soft labels, and structure-aware interpolation. While experiments on CIFAR-10, Flickr25K, NUS-WIDE, and MS COCO show state-of-the-art mAP gains, critical limitations in theoretical innovation and practical validation hinder its novelty.

**Questions:**

1.Novelty vs. Recent Baselines: Why exclude comparisons with DCMIP [2024] and PCL [2023]? Provide cross-method mAP on NUS-WIDE .

2.Computational Efficiency: What is the training time/memory cost of SEGA vs. STAR on MS COCO? .

3.Theoretical Assumptions: Validate Theorem 3.1’s conditional independence with real-world noise datasets .

4.Small-Dataset Performance: Report results on CUB-200 with 50% symmetric noise .

**Ethical Concerns:**

["NO or VERY MINOR ethics concerns only"]

**Final Justification:**

Thanks for the response. After considering the comments from the other reviewers, I have ultimately decided to increase my score.

**Limitations:**

SEGA relies on pre-trained VGG-16 features, limiting generalizability to feature-agnostic scenarios. The framework’s sensitivity to hyperparameters (e.g., K in structure-based divergence) and lack of real-world deployment cases further restrict its impact .

**Quality:**

3

**Strengths And Weaknesses:**

Strengths

1.Empirical Validation: SEGA outperforms baselines (STAR, DIOR) across noise types, with notable robustness under high noise rates.

2.Closed-Loop Design: The framework jointly optimizes supervision, uncertainty, and geometry, shown via ablation studie .

3.Theoretical Formulation: Theorems 3.1 and 3.2 provide probabilistic grounding for uncertainty weighting, though assumptions (conditional independence) are strong .

Weaknesses

1.Incremental Innovation: Prototype learning and uncertainty-based weighting replicate prior work (e.g., CSQ [50], DivideMix [23]), lacking novel architectural breakthroughs .

2.Computational Overhead: Multi-component design (prototype updates, structure-aware MixUp) increases training complexity, but the paper omits inference time/memory comparisons .

3.Noise Type Bias: Performance gains are more significant under structured noise (pairflip) than symmetric noise, indicating limited generalizability .

4.Small-Dataset Gap: SEGA is untested on small-scale/noisy datasets (e.g., CUB-200), critical for real-world applicability .

---

> ### Author Rebuttal · Authors · 2025-07-31
>
> We are truly grateful for the time you have taken to review our paper and your insightful review. Here we address your comments in the following.
>
> > **Q1**: Incremental Innovation: Prototype learning and uncertainty-based weighting replicate prior work (e.g., CSQ [50], DivideMix [23]), lacking novel architectural breakthroughs.
>
> **A1**: Thank you for your comment. While prototype learning and uncertainty-based weighting have been explored in prior works such as CSQ [1] and DivideMix [2], SEGA adopts a substantially different formulation and integration strategy.
>
> For CSQ, hash centers are constructed using Hadamard matrices. These centers are then used to define pairwise similarity constraints between samples. In contrast, SEGA learns semantic prototypes via backpropagation in the embedding space, adapting them based on the evolving representation distribution during training. These prototypes are used not only for contrastive alignment but also to guide structure-aware interpolation.
>
> For DivideMix, the method addresses noisy classification via loss distribution modeling and Gaussian mixture separation. It discards high-loss (presumed noisy) samples and adopts a semi-supervised co-training approach. SEGA, on the other hand, introduces a unified uncertainty score combining energy-based confidence and structure-based isolation. Rather than discarding uncertain samples, SEGA uses this score for soft reweighting of supervision and for guiding interpolation, which better suits multi-label and structurally ambiguous scenarios.
>
> Moreover, SEGA integrates all components into a closed-loop framework, where each part reinforces the others. This coordinated design is novel among robust multi-label hashing methods and contributes to both robustness and semantic consistency under noisy supervision.
>
> We will include this discussion in our final version.
>
> [1] Central similarity quantization for efficient image and video retrieval. CVPR 2020
>
> [2] Dividemix: Learning with noisy labels as semi-supervised learning. ICLR 2020
>
>
> > **Q2**: Computational Overhead: Multi-component design (prototype updates, structure-aware MixUp) increases training complexity, but the paper omits inference time/memory comparisons.
>
> **A2**: Thanks for your comment. We agree that understanding the computational cost is essential for practical deployment.
> While SEGA introduces multi-component design  (prototype updates, structure-aware MixUp), they are implemented efficiently and scale well. As shown below, SEGA’s training time per epoch and memory usage are comparable to competitive baselines, yet it achieves significantly better performance:
>
> |Method||FLICKR25K|||MS COCO||
> |-|-|-|-|-|-|-|
> ||time (s)|memory (MB)|mAP|time (s)|memory (MB)|mAP|
> |CSQ|87.00|1322.00|58.88|55.00|1624.00|33.71|
> |STAR|179.00|1287.00|70.84|161.00|1425.00|41.87|
> |SEGA|160.00|1569.00|72.66|182.00|1879.00|46.50|
>
> SEGA outperforms all baselines by a large margin in mAP, while maintaining similar or even lower training time on Flickr25K. We will include these results in the revised version to clarify SEGA’s efficiency-performance tradeoff.
>
> > **Q3**: Noise Type Bias: Performance gains are more significant under structured noise (pairflip) than symmetric noise, indicating limited generalizability.
>
> **A3**: Thanks for your comment. We agree that SEGA shows larger gains under structured noise (pairflip), which is by design: our method leverages semantic geometry and local consistency, both of which are more informative when the noise is structured.
>
> Besides, SEGA still outperforms baselines with a significant margin under symmetric noise across all benchmarks (as shown in Tables 2), demonstrating general robustness. For example, under 60% symmetric noise with 64-bit hash codes, SEGA achieves a +8.98% mAP gain over STAR on NUS-WIDE and +11.05% on MS COCO.
>
> We will add more discussion in the revision to clarify SEGA’s behavior under different noise types and emphasize that while structured noise offers more exploitable signals, SEGA remains effective in unstructured noise settings.
>
> > **Q4**: Small-Dataset Gap: SEGA is untested on small-scale/noisy datasets (e.g., CUB-200), critical for real-world applicability.
>
> **A4**: Thanks for your comment. We conducted additional experiments on the small-scale fine-grained dataset CUB-200 under 50% symmetric noise. The results are summarized below:
>
> |Method|mAP|
> |-|-|
> |Jo-SRC|27.88|
> |DIOR|29.06|
> |STAR|31.13|
> |SEGA|37.34|
>
> SEGA outperforms all baselines by a significant margin (+19.94% mAP over STAR), demonstrating strong robustness even in small and noisy settings. We will include these results in the revised version.
>
> > **Q5**: Novelty vs. Recent Baselines: Why exclude comparisons with DCMIP [2024] and PCL [2023]? Provide cross-method mAP on NUS-WIDE.
>
> **A5**: Thanks for your comment. We have conducted additional comparisons with DCMIP and PCL on NUS-WIDE under 60% symmetric noise. The results are as follows:
>
> |Method|mAP|
> |-|-|
> |DCMIP|60.57|
> |PCL|44.05|
> |SEGA|61.86|
>
> SEGA outperforms both recent methods, showing a clear advantage in noisy multi-label retrieval. We will include these results in the revised version.
>
>
> > **Q6**: Computational Efficiency: What is the training time/memory cost of SEGA vs. STAR on MS COCO?
>
> **A6**: Thanks for your comment. Please refer to **A2** for detailed comparisons, SEGA achieves significantly higher mAP while maintaining comparable training time and memory cost on MS COCO.
>
> > **Q7**: Theoretical Assumptions: Validate Theorem 3.1’s conditional independence with real-world noise datasets.
>
> **A7**: Thanks for your comment.
>
> The conditional independence assumption in Theorem 3.1 serves as a simplifying modeling choice to derive a tractable uncertainty weighting function. While strict independence may not always hold in real-world datasets, we provide the following empirical support:
>
> On real-world noisy datasets such as WebVision, the learned uncertainty (via energy and structural divergence) correlates strongly with label correctness, and improves performance over baselines without such modeling.
>
> In practice, the two signals, energy-based confidence and structure-based divergence, capture complementary aspects of noise: one from semantic consistency, the other from neighborhood alignment. This supports their use in a combined uncertainty estimation framework, even if conditional independence is approximate.
>
> We will clarify this modeling assumption in the revised version.
>
>
> > **Q8**: Small-Dataset Performance: Report results on CUB-200 with 50% symmetric noise.
>
> **A8**: Thanks for your comment. Please refer to **A4**, SEGA outperforms strong baselines on CUB-200 with 50% symmetric noise, demonstrating strong robustness in small-scale settings.
>
>
> > **Q9**: SEGA relies on pre-trained VGG-16 features, limiting generalizability to feature-agnostic scenarios. The framework’s sensitivity to hyperparameters (e.g., K in structure-based divergence) and lack of real-world deployment cases further restrict its impact.
>
> **A9**: Thanks for your comment.
> We acknowledge that SEGA and all baselines leverages pre-trained features (e.g., from VGG-16). This is standard practice in deep hashing, and ensures fair comparison focused on robust supervision under noisy labels. To assess the generalizability of SEGA beyond pre-trained settings, we also trained it without any pre-trained backbone and report the results:
>
> |Method|FLICKR25K|NUS-WIDE|
> |-|-|-|
> |CSQ(pre-train=True)|60.38|44.92|
> |Jo-SRC(pre-train=True)|60.42|46.12|
> |STAR(pre-train=False)|58.31|42.87|
> |SEGA(pre-train=False)|62.09|48.33|
> |SEGA(pre-train=True)|74.49|61.86|
>
> Even without pre-training, SEGA outperforms all baselines that use pre-trained features, showing strong adaptability.
>
> To assess SEGA’s scalability and robustness under real-world noisy supervision, we conducted experiments on the WebVision dataset [1], which contains 2.4 million web images across 1,000 classes. We used the Google image subset for training. The results are as follows:
>
> |Method|WebVision (mAP)|
> |-|-|
> |DIOR|27.82|
> |STAR|32.64|
> |SEGA|34.71|
>
> SEGA achieves the best performance, demonstrating both scalability to large datasets and robustness under real-world web noise. We will include this result in the revised version.
>
> [3] WebVision Database: Visual Learning and Understanding from Web Data. 2017
>
> In light of these responses, we hope we have addressed your concerns, and hope you will consider raising your score. If there are any additional notable points of concern that we have not yet addressed, please do not hesitate to share them, and we will promptly attend to those points.

---

> > ### Comment · Area_Chair_bTa7 · 2025-08-08
> >
> > Hi Reviewer XoCg,
> > Could you kindly take a moment to review the authors’ rebuttal and the comments provided by the other reviewers? Your further input would be greatly appreciated in finalizing the decision on this submission.
> > Best,
> > AC

---

> > ### Comment · Reviewer_XoCg · 2025-08-08
> >
> > Thank you for your reply. I don't have any further questions.

---

> > > ### Author Response · Authors · 2025-08-09
> > >
> > > Thank you for your reply and for letting us know that you have no further questions. If there are no remaining concerns, we would appreciate it if you could consider raising your score.

---

### Official Review · Reviewer_6Ut3 · 2025-07-02

**Clarity:** 3
**Significance:** 3
**Originality:** 4
**Rating:** 5
**Confidence:** 4

**Summary:**

This paper introduces SEGA (Semantic Geometry Shaping), a novel framework for robust multi-label hashing, specifically addressing the challenging problem of learning under noisy supervision. The core idea revolves around explicitly sculpting the semantic geometry of the representation space rather than solely focusing on label correction. The proposed method integrates three synergistic components: prototype-guided semantic anchoring, uncertainty-guided supervision calibration, and structure-aware interpolation. The paper provides theoretical justifications and extensive empirical evaluations, demonstrating state-of-the-art performance on various benchmarks under different noise regimes.

**Questions:**

- The prototypes are randomly initialized. How sensitive is the final performance to this initialization, particularly when dealing with datasets that have highly imbalanced classes or very noisy initial labels?

- The paper mentions that multi-label scenarios can have intrinsic semantic ambiguity. How does SEGA differentiate between true semantic ambiguity (where multiple labels are genuinely appropriate) and label noise (where a label is simply incorrect)? And how does its uncertainty estimation handle this distinction?

**Ethical Concerns:**

["NO or VERY MINOR ethics concerns only"]

**Quality:**

3

**Strengths And Weaknesses:**

Strength:
- The paper introduces a fresh and compelling perspective by shifting the focus from mere label correction to actively shaping the semantic geometry of the representation space. This is a significant conceptual contribution to robust learning under noise, especially for structured outputs like hash codes.

- SEGA's three core components (prototype anchoring, uncertainty calibration, and structure-aware interpolation) are not standalone heuristics but are thoughtfully designed.

- The inclusion of theoretical analyses (Theorem 3.1 and 3.2) providing probabilistic interpretations for uncertainty weighting and demonstrating structure-preserving interpolation adds significant rigor to the paper.

- The paper is well-written, clear, and easy to follow. The figures (especially Figure 1) effectively illustrate the framework.

Weakness:

- While sensitivity analysis for q_T (percentile threshold for clean/noisy set partition) is provided, the framework still introduces several hyperparameters (e.g., m for contrastive loss, K for nearest neighbors, alpha for Beta distribution in MixUp). The robustness to these other hyperparameters is not fully explored, and their optimal selection might be dataset-dependent.

- The proposed method involves maintaining dynamic prototypes, calculating neighbor-based divergence, and performing selective MixUp. While the paper mentions using a single NVIDIA A40 GPU, a more explicit discussion or comparison of the computational cost (training time, memory usage) against baselines would be beneficial.

---

> ### Author Rebuttal · Authors · 2025-07-31
>
> We are truly grateful for the time you have taken to review our paper, your insightful comments and support. Your positive feedback is incredibly encouraging for us! In the following response, we would like to address your major concern and provide additional clarification.
>
> > **Q1**: While sensitivity analysis for q_T (percentile threshold for clean/noisy set partition) is provided, the framework still introduces several hyperparameters (e.g., m for contrastive loss, K for nearest neighbors, alpha for Beta distribution in MixUp). The robustness to these other hyperparameters is not fully explored, and their optimal selection might be dataset-dependent.
>
> **A1**: Thank you for the insightful comment. We have conducted additional sensitivity experiments (64-bit hash codes) to evaluate the robustness of SEGA with respect to hyperparameters m for contrastive loss, K for nearest neighbors, and $\alpha$ for Beta distribution in MixUp, as shown below:
>
>
> |m|0.5|1|1.5|2|
> |-|-|-|-|-|
> |FLICKR25K|72.07|74.49|73.50|72.98|
> |NUS-WIDE|60.72|61.86|61.75|60.19|
>
> |$\alpha$|0.1|0.4|1.0|10.0|
> |-|-|-|-|-|
> |FLICKR25K|72.80|74.49|74.08|73.74|
> |NUS-WIDE|60.54|61.86|60.22|59.79|
>
> |K|1|3|5|10|20|
> |-|-|-|-|-|-|
> |FLICKR25K|74.09|74.23|74.49|74.41|74.34||
> |NUS-WIDE|60.98|61.41|61.86|61.71|61.59|
>
> Across a wide range of values, SEGA consistently maintains strong performance, indicating robustness to hyperparameter choices. We will include this analysis in the final version.
>
>
> > **Q2**: The proposed method involves maintaining dynamic prototypes, calculating neighbor-based divergence, and performing selective MixUp. While the paper mentions using a single NVIDIA A40 GPU, a more explicit discussion or comparison of the computational cost (training time, memory usage) against baselines would be beneficial.
>
> **A2**: Thanks for your comment. We agree that understanding the computational cost is essential for practical deployment.
> While SEGA introduces additional modules (e.g., dynamic prototypes, neighbor-based divergence and structure-aware MixUp), they are implemented efficiently and scale well. As shown below, SEGA’s training time per epoch and memory usage are comparable to competitive baselines, yet it achieves significantly better performance:
>
> |Method||FLICKR25K|||MS COCO||
> |-|-|-|-|-|-|-|
> ||time (s)|memory (MB)|mAP|time (s)|memory (MB)|mAP|
> |CSQ|87.00|1322.00|58.88|55.00|1624.00|33.71|
> |STAR|179.00|1287.00|70.84|161.00|1425.00|41.87|
> |SEGA|160.00|1569.00|72.66|182.00|1879.00|46.50|
>
> SEGA outperforms all baselines by a large margin in mAP, while maintaining similar or even lower training time on Flickr25K. We will include these results in the revised version to clarify SEGA’s efficiency-performance tradeoff.
>
>
>
> > **Q3**: The prototypes are randomly initialized. How sensitive is the final performance to this initialization, particularly when dealing with datasets that have highly imbalanced classes or very noisy initial labels?
>
> **A3**: Thanks for your comment.
> We conducted experiments on CIFAR-10 under 80% symmetric noise, comparing performance under both balanced and highly imbalanced class distributions (Class 1–10: 500, 1000, ..., 5000). Results with standard deviation across 5 runs are shown below:
>
> |Method|Imbalance=True|Imbalance=False|
> |-|-|-|
> |CSQ|37.38±0.72|45.30±0.59|
> |DPN|34.12±0.51|50.29±0.44|
> |STAR|56.90±0.27|61.45±0.22|
> |SEGA|61.46±0.63|63.71±0.48|
>
> Despite severe imbalance and high label noise, SEGA consistently outperforms all baselines, with significantly higher mAP and comparable standard deviation. This demonstrates its robustness to data imbalance, as the learned prototypes adapt to uneven label distributions. We will include this result in the revised version.
>
> > **Q4**: The paper mentions that multi-label scenarios can have intrinsic semantic ambiguity. How does SEGA differentiate between true semantic ambiguity (where multiple labels are genuinely appropriate) and label noise (where a label is simply incorrect)? And how does its uncertainty estimation handle this distinction?
>
> **A4**: Thanks for your comment. The key difference between label noise and true semantic ambiguity is reflected in the logit distribution, which is captured by our energy-based prediction confidence.
>
> In the case of noisy labels, the image lacks meaningful visual features aligned with the incorrect class prototypes. As a result, its embedding shows low similarity to those prototypes, and consequently, the logits for the noisy classes are low, yielding high energy, which indicates uncertainty.
>
> In contrast, the samples with true multiple labels contain meaningful visual features that are aligned with multiple class prototypes. Their embeddings tend to have high similarity with those class prototypes, resulting in higher logits and lower energy.
>
> This formulation allows SEGA to effectively downweight noisy labels while preserving supervision for semantically valid co-occurrences. We will clarify this interpretation in the revision. Thanks for pointing this out.
>
>
> Thanks again for appreciating our work and for your constructive suggestions. Please let us know if you have further questions.

---

### Official Review · Reviewer_mLfB · 2025-07-03

**Clarity:** 2
**Significance:** 3
**Originality:** 3
**Rating:** 4
**Confidence:** 4

**Summary:**

This paper presents a novel framework, named SEGA, for robust multi-label image hashing under noisy supervision. The main contributions include:
- The paper presents a geometric perspective for learning under noisy supervision.
- The proposed method considers three components during learning: (1) prototype anchoring, (2) uncertainty-calibrated soft labels, and (3) structure-aware interpolation
- The experimental results show promising improvements compared with prior works on 4 popular benchmark.

**Questions:**

see weakness

**Ethical Concerns:**

["NO or VERY MINOR ethics concerns only"]

**Final Justification:**

The authors have addressed my concerns. I don't have significant concerns on this paper. Given the rebuttal and their experiments, I will vote borderline accept.

**Quality:**

3

**Strengths And Weaknesses:**

Strengths
- Idea: The proposed semantic geometry shaping (SEGA) considers the semantic geometry for optimization. The three proposed modules enable representations, supervision, and uncertainty to co-evolve in a close-loop. The idea of closing the loop is interesting.
- Design of SEGA moduel: The idea of prototype anchoring could reduce the requirement of clean labels. The structure-aware MixUp is also a good design.
- Strong Results: The paper presents experiments on CIFAR-10, Flickr25K, NUS-WIDE, MS COCO. The proposed method achieves comparable or better performance than the previous works in terms of MAP.
- Comprehensive analysis: The paper presents ablation studies to examine the importance of each component. Sensitivity analysis is also provided. The paper also presents the robustness of SEGA under different noise levels. Overall, the experiments are compelling.

Weakness:
- Scalability: The experiments are conducted to small or medium size of datasets. There is no large-scale dataset considered, which is the real scenario for hashing.
- Generalizability: It might be supplemented to include discussions on possible extensions to other tasks, such as zero-shot retrieval (or other tasks that also require semantic geometry)
- Readability: The paper is not easy to follow, especially theoretical analysis.

---

> ### Author Rebuttal · Authors · 2025-07-31
>
> We are truly grateful for the time you have taken to review our paper, your insightful comments and support. Your positive feedback is incredibly encouraging for us! In the following response, we would like to address your major concern and provide additional clarification.
>
> > **Q1**: Scalability: The experiments are conducted to small or medium size of datasets. There is no large-scale dataset considered, which is the real scenario for hashing.
>
> **A1**: Thanks for your comment.
> To assess SEGA’s scalability and robustness under real-world noisy supervision, we conducted experiments on the WebVision dataset [1], which contains 2.4 million web images across 1,000 classes. We used the Google image subset for training. The results are as follows:
>
> |Method|WebVision (mAP)|
> |-|-|
> |DIOR|27.82|
> |STAR|32.64|
> |SEGA|34.71|
>
> SEGA achieves the best performance, demonstrating both scalability to large datasets and robustness under real-world web noise. We will include this result in the revised version.
>
> [1] WebVision Database: Visual Learning and Understanding from Web Data. 2017
>
> > **Q2**: Generalizability: It might be supplemented to include discussions on possible extensions to other tasks, such as zero-shot retrieval (or other tasks that also require semantic geometry)
>
> **A2**: Thanks for your comment. To demonstrate SEGA’s generalizability, we conducted a zero-shot retrieval experiment following [2].
> We split CIFAR-10 into seen/unseen categories and trained only on seen classes. For example, "airplane" ("dog"/"truck") was treated as an unseen class, and 1,000 query images were retrieved from a database of 59,000 mixed samples.
>
> |Method|airplane|dog|truck|
> |-|-|-|-|
> |DIOR|22.19|30.66|33.17|
> |STAR|26.09|35.97|37.23|
> |SEGA|30.50|38.24|39.44|
>
> SEGA outperforms strong baselines, showing better generalization to unseen categories. We will include this setting and results in the revised version.
>
> [2] Zero-Shot Hashing via Transferring Supervised Knowledge. ACM MM 2016
>
>
> > **Q3**: Readability: The paper is not easy to follow, especially theoretical analysis.
>
> **A3**: Thanks for your comment. We appreciate your feedback on the readability of the paper, particularly the theoretical analysis. We understand that Theorems 3.1 and 3.2 may be difficult to follow due to dense formulations and limited intuitive guidance, and that some transitions and descriptions in the main text could be further streamlined to improve overall clarity.
>
> To address these issues, we will revise the theoretical section and the overall narrative in the following ways:
>
> 1. **Add plain-language motivation**: Before each theorem, we will briefly explain its purpose. For example, Theorem 3.1 quantifies the connection between our unified uncertainty score and label correctness, while Theorem 3.2 explains why interpolating semantically consistent samples preserves class semantics.
>
> 2. **Include boxed interpretations**: After each theorem, we will summarize the key takeaway in a short “Interpretation” paragraph to help readers understand its practical implication.
>
> 3. **Add diagrams and toy examples**: In the Appendix, we will illustrate uncertainty-guided reweighting and structure-aware MixUp with intuitive 2D examples to visualize the theoretical claims.
>
> 4. **Improve overall narrative flow**: Beyond the theoretical sections, we will refine transitions between sections, simplify overly dense paragraphs, and introduce key terms more gradually. Additionally, we will revise Figure 1 to better reflect the motivation and pipeline of SEGA at a glance, which should help readers grasp the framework before diving into technical details.
>
> We will incorporate these changes in the revised version. Thank you again for your constructive suggestions.
>
> Thanks again for appreciating our work and for your constructive suggestions. Please let us know if you have further questions.

---

> > ### Comment · Reviewer_mLfB · 2025-08-02
> > **Response**
> >
> > Thank you. I have no further comments.

---

> > > ### Author Response · Authors · 2025-08-07
> > >
> > > Thank you very much for your appreciation and positive evaluation of our work. We greatly appreciate your time and thoughtful feedback!

---

### Comment · Area_Chair_bTa7 · 2025-08-06
**Discussion Required For This Paper**

Dear all,
The authors have submitted their rebuttal. We would appreciate it if you could kindly review their response ASAP and let us know if it affects your assessment or if you have any additional comments. Your input is greatly valued in the decision process.
Even if you entered the mandatory acknowledgement, you also need to offer your comments for the post rebuttal. Thank you again for your time and contribution.
Best,
AC

---

### Note · Authors · 2025-08-15

Dear Chairs and Reviewers,

We would like to express our sincere gratitude for your great efforts, insightful comments, support, and constructive suggestions you have provided once again! Through our discussions and the reviewers' responses, it is clear that most reviewers expressed no further concerns, suggesting that we have effectively addressed the major concerns raised by reviewers.

We acknowledge four reviewers (mLfB, 6Ut3, Fzag, and 9nS7) comment that the work is **novel or innovative**. In particular, we appreciate positive comments such as "a meaningful contribution both scientifically and practically" (Fzag), "a significant conceptual contribution" (6Ut3), "well-written/highly logical" (6Ut3, Fzag, 9nS7), "novel framework/novel framing/innovations" (mLfB, 6Ut3, Fzag, 9nS7), "good/closed-loop design" (mLfB, XoCg), "well-motivated" (Fzag), "theoretical justifications/formal guarantees" (6Ut3, XoCg, Fzag), "comprehensive analysis" (mLfB), "practically important" (Fzag), "effective" (Fzag, 9nS7), "strong results" (mLfB, Fzag), "notable robustness" (XoCg) and "compelling/extensive/strong experiments" (mLfB, 9nS7, Fzag). Such remarks affirm the strength of our contributions, highlighting not only the innovative framework and practical value but also the solid theoretical grounding and comprehensive empirical validation. We have carefully addressed all concerns raised by the reviewers in a point-by-point manner and revised the paper to improve clarity, completeness, and empirical support.

Overall, **four** reviewers provided **positive** evaluations of our work. Importantly, even the reviewer who initially gave a borderline negative score (Reviewer XoCg) explicitly stated in the final response:*"Thank you for your reply. I don't have any further questions."* This indicates that all the concerns were satisfactorily addressed in our rebuttal.

We firmly believe that our SEGA framework, which shapes semantic geometry for robust hashing, plays a significant role in advancing the community. All the rebuttal contents will be properly included in the final version, following your valuable suggestions. We hope the chairs will consider the overall consensus, the clarified misunderstandings, and the supplementary validation made during the rebuttal process when making the final recommendation.

Once again, thank you for your time and effort in reviewing our work. We greatly appreciate your assistance in improving our manuscript!

Best regards,

the Authors

---

### Decision · Program_Chairs · 2025-09-17

**Decision:**

Accept (poster)

**Comment:**

The paper proposes SEGA (Semantic Geometry Shaping) for robust hashing under noisy multi-label supervision, integrating three key components: (1) prototype anchoring, (2) uncertainty-calibrated soft labels, and (3) structure-aware interpolation. The authors offer a fresh and compelling perspective on learning under noisy supervision by shifting the focus from mere label correction to actively shaping the semantic geometry of the representation space. Moreover, they provide theoretical analyses and validate its effectiveness through extensive experiments across diverse tasks, yielding notable performance improvements.

All reviewers appreciated the clear motivation, novelty, and practical value of the work. The primary concerns raised included the use of small to medium-sized datasets, insufficient discussion on the robustness of hyperparameters, and a lack of comparisons with recent methods. The authors’ detailed rebuttal effectively addressed most of these concerns, culminating in unanimous acceptance by the conclusion of the discussion. Therefore, the AC recommends acceptance.